# The Nariño Cat, the Tigrinas and Their Problematic Systematics and Phylogeography: The Real Story

**DOI:** 10.3390/ani15131891

**Published:** 2025-06-26

**Authors:** Manuel Ruiz-García, Javier Vega, Myreya Pinedo-Castro, Joseph Mark Shostell

**Affiliations:** 1Laboratorio de Genética de Poblaciones Molecular-Biología Evolutiva, Unidad de Genética, Departamento de Biología, Facultad de Ciencias, Pontificia Universidad Javeriana, Bogotá 110311, Colombia; vegagjdario@javeriana.edu.co (J.V.); pinedom@javeriana.edu.co (M.P.-C.); 2Department of Math, Science and Technology, University of Minnesota Crookston, Crookston, MN 56716, USA; joseph.shostell@gmail.com

**Keywords:** Andean Colombia, hybridization, introgression, *Leopardus*, mitogenomes, Nariño cat, *ND5* gene, neotropics, phylogeny, phylogeography, tigrinas

## Abstract

In the last 20 years, some new species of neotropical tigrinas have been claimed (*guttulus*, *emiliae*, *pardinoides*, *narinensis*). Probably, the most controversial of them has been that of the Nariño cat (*Leopardus narinensis*) since only a single specimen of this presumed new taxon has been analyzed. For this reason, we have carried out the most extensive analysis of the Andean tigrina with mitochondrial DNA to date, including some samples taken from this single specimen across different years to see if the genetic results of this specimen were consistent. Mitochondrial analyses (with two databases, mt*ND5* gene and complete mitogenomes) revealed inconsistent results since the phylogenetic relationships of the Nariño cat with other tigrinas and with all species of the *Leopardus* genus were variable. The oldest samples seem to be the ones that show the most robust results in reference to the phylogenetic position occupied by the Nariño cat, showing it as a new taxon. However, this molecular study shows that the Andean tigrina (*L. pardinoides*) is made up of different polyphyletic taxa, especially in Colombia and Ecuador, although this issue is complex because the reproductive barrier mechanisms between many species of the *Leopardus* genus are weak and their diversification has been recent, which has allowed numerous events of incomplete linage sorting, ancestral introgression, and recent hybridization within this genus. Another relevant result of this study is that to fully understand the systematics of tigrinas, it is necessary to analyze as many specimens with the most complete geographical distribution as possible. Nevertheless, the most realistic method to try to understand and solve the evolutionary and systematic problem of tigrinas is to carry out in-depth studies with Whole Genome Sequencing (WGS).

## 1. Introduction

Wild cats have always attracted the attention of humans for their esthetics and their abilities as predators. They are often of extreme importance in food chains and are emblematic and iconic species in issues related to biological conservation [1,2,3].

In Neotropics (Central and South America), there are currently three lineages of felines. They are the Panthera lineage, represented by the jaguar (*Panthera onca*), the Puma lineage, represented by two species, the puma (*Puma concolor*) and the jaguarundi (*Puma* (=*Herpailurus*) *yagouaroundi*), and the Ocelot lineage, whose number of species is in discussion. Some authors suggest that there are eight species within the Ocelot lineage [4]: the Andean mountain cat (*Leopardus jacobita*); the ocelot (*Leopardus pardalis*); the margay (*Leopardus wiedii*); the Pampas cat (*Leopardus colocola*); the tigrina, oncilla, or tiger-cat (*Leopardus tigrinus*); the southern tigrina (*Leopardus guttulus*; previously classified into the “traditional” tigrina); Geoffroy’s cat (*Leopardus geoffroyi*); and the guiña or kodkod (*Leopardus guigna*). More recently, some authors have proposed the existence of additional species within the Ocelot lineage [5,6,7,8]. Nascimento et al. [6] divided the “traditional” Pampas cat (*L. colocola*) into five different species: *L. colocola* “sensu stricto”, *L. pajeros*, *L. garleppi*, *L. braccatus*, and *L. munoai*. Twenty-seven years earlier, García-Perea [9] stated that there were three species and 10 subspecies of Pampas cats (*L. colocola*, *L. pajeros*, and *L. braccatus*). The claim was based off an analysis of skull and coat characteristics of 86 specimens. Unlike previous authors, we are reluctant to divide the Pampas cat into different species for two reasons: (1) These “species” form a homogeneous and monophyletic clade based off individual mt genes and mitogenomes. The genetic distances among the different groups of Pampas cats in this clade are smaller than the genetic distances for any *Leopardus* species pair. (2) Based off the analysis of DNA microsatellites, the levels of gene flow among Pampas cat group pairs were substantially higher than for any *Leopardus* species pairs [10,11]. However, we suggest a focus on the tigrinas, where additional species will likely occur.

Traditionally, four subspecies within the tigrina [12,13,14] have been defined until recently: (1) *L. t. oncilla* in Costa Rica and Panama and may be present possibly as far north as Nicaragua; (2) *L. t. pardinoides*, in the Andean zone from western Venezuela, Colombia, Ecuador and northern Peru; (3) *L. t. tigrinus* from eastern Venezuela, Guianas and northeastern Brazil; and (4) *L. t. guttulus* in central and southern Brazil, Paraguay and northeastern Argentina (Misiones province).

However, surprising discoveries occurred when southern Brazilian tigrinas were analyzed [15]. Using mt*ND5* gene sequences and nine microsatellites, Trigo et al. [15] detected and characterized a hybrid zone between Geoffroy’s cat and the tigrina in southern Brazil. Both species are mostly allopatric throughout their ranges with a narrow contact zone that includes this Brazilian area. Trigo et al. [15] detected at least 14 individuals (most of them originating from the geographical contact zone) exhibiting signs of interspecific genomic introgression of *L. geoffroyi* within the southern Brazilian tigrina. Later, Trigo et al. [16] expanded the analysis of the hybridization among Geoffroy’s cat, the Pampas cat, and the southern Brazilian tigrina. For the first time, authors have provided strong evidence of ancient hybridization and introgression between the Pampas cat and populations of tigrina in northeastern Brazil (*L. tigrinus emiliae*). In contrast, the southern Brazilian tigrina has engaged in recent and continuing hybridizations with Geoffroy’s cat. These have led to extreme levels of interspecific admixture along their contact zone. They demonstrated that two seemingly continuous Brazilian tigrina populations (*L. guttulus* and *L. t. emiliae*) did not show any evidence of gene flow [16]. The lack of evidence could support the idea that the two Brazilian tigrina populations as two separate species: *L. tigrinus* in northeastern Brazil and *L. guttulus* in southern and central Brazil.

Sunquist and Sunquist [2] stated, “Scientists anticipate that as more genetic information becomes available from other parts of the cat’s distribution (referred to the tigrina), we may see a multitude of oncilla species.” Effectively, this was what happened.

Trigo et al. [17] expanded their molecular genetic analyses, as well as some morphological data, of *L. guttulus* and *L. geoffroyi* in southern Brazil. Despite a high level of hybridization, samples collected in southern Brazil formed into two recognizable clusters (genetically and morphologically). These two clusters corresponded to ‘pure’ individuals of each species. These ‘pure’ individuals were mainly sampled in regions further from the contact zone. But hybrids concentrated at the interface between the two biomes. In agreement with the molecular results, the morphological data revealed a strong spatial structure but displayed an even more marked separation between the two ‘pure’ clusters. Hybrids often did not present intermediate body sizes and could not be clearly distinguished morphologically from the parental forms. This could be related to selective pressure acting on the hybrids, limiting their dispersal away from the hybrid zone. The limitation would favor genomic combinations that maintain adaptive phenotypic features of one or the other parental species in their respective biomes.

Li et al. [18] showed signs of phylogenetic discordance that were abundant in the genomes of modern cats. Hybridization and introgression were indicated as the most likely causes in most of these cases. The wild cats with the greatest amount of phylogenetic discordance were within the Ocelot lineage. This study showed that the *L. tigrinus* from northeastern Brazil possessed *L. colocola* mtDNA with a tigrina nuclear DNA background, as previously observed. Additionally, it was suggested that the northeastern Brazilian tigrina constituted a full species, the Ceará tigrina (*Leopardus emiliae*) [5]. However, based on the analysis of the complete genome, Lescroart et al. [19] still considered it a subspecies of *L. tigrinus* (*L. t. emiliae*).

But the situation became even more complex when some tigrinas in the Andean and Pacific regions of northwestern South America began to be molecularly analyzed. A complex situation was found in the geographical distribution area of the traditional and putative subspecies, *L. t. pardinoides*, in the Andean and inter-Andean areas of Venezuela, Colombia, Ecuador and Peru [7]. Different tigrina groups were detected when two mt datasets were analyzed (mt*ATP8* ± mt*16S rRNA*; mitogenomes): (1) The tigrina from Central America and the trans-Andean areas of Colombia and Ecuador was differentiable from other tigrina lineages. This was the tigrina taxon more related to *L. geoffroyi* and *L. guigna*. One of the most exciting discoveries was that the traditional *L. t. oncilla* was not only distributed in Costa Rica (Central America), but it was also distributed in the trans Andean area (Pacific slopes) of Colombia and Ecuador [7]. They detected one exemplar in the Valle del Cauca department in Colombia and another in Intag (Imbabura province) in Ecuador associated with two Costa Rican tigrinas. (2) Another tigrina lineage consisted of three individuals, one from Lamas (San Martın department) in Peru and another two with Colombian origins (one from Chingaza NP, Cundinamarca department, and another from the Tolima department). (3) Another tigrina lineage consisted of individuals from different Andean departments of Colombia (Quindıo, Boyaca and Huila) and one individual from the Cotopaxi province in Ecuador. (4) However, the majority of the Andean tigrinas studied by Ruiz-García et al. [7] shared haplotypes very similar to those found in margays and in ocelots. Tigrinas with haplotypes closely related with margays and ocelots were detected in the Andean areas of Venezuela, Colombia, Ecuador, Bolivia and northwestern Argentina. For example, when just mt*ATP8*-mt*16S rRNA* haplotypes were considered, some of the tigrinas shared the same haplotypes found in margays, whereas other tigrinas shared the same haplotypes as that of ocelots. This was the first time that possible historical introgression and/or hybridization (depending on each individual case) of ocelot–margay was detected inside *L. tigrinus.* Later, Ramirez et al. [20], using ABBA-BABBA tests [21], and Lescroart et al. [19] confirmed an ancient introgression of the ocelot (or an ancestor of the ocelot) within the ancestors of the northern tigrinas as well as of Geoffroyi’s cat. (5) Ruiz-García et al. [7] detected one specimen with a strange phenotype as well as with a surprising mitochondrial relationship with other *Leopardus* taxa. In that work, this specimen did not receive any specific name. Some years later, Ruiz-García et al. [8] published a report of this specimen, which caused some controversy. It was the case of the Nariño cat (*Leopardus narinensis*), a specimen from the Nariño department in the southern Andean area of Colombia.

In the current paper, we share the most recent molecular analyses of the Nariño cat. First, we describe its discovery.

In 2000, eminent Colombian biologist Dr. Jorge Ignacio Hernández-Camacho (alias Mono Hernández) and Dr. Manuel Ruiz-García (lead author of the current paper) met and discussed the different mammalian species of the Colombian fauna. One of the most interesting revelations made by Hernández-Camacho was that in the 1980s, he had seen a “strange” small spotted cat in the Cali Zoo that he had not been able to classify to the species level. This was surprising because Hernández-Camacho’s research focus was the neotropics and he had an exhaustive knowledge of Neotropical fauna and flora. He also commented that sometime later he had met with the director of the Cali Zoo and asked if the zoo would contact him when this unidentifiable feline died. He wanted to preserve the skin and the skeleton of this unusual wild cat. Unfortunately, the director of the Cali Zoo told him that the feline had died, and it had been cremated. Hernández-Camacho had provided other information of this unusual cat. During the 1970s, he had received several reports from peasants from southern Colombia of a small wild cat that had some different physical features from the “traditional” tigrina that inhabited higher elevations. He also informed Dr. Ruiz-García that on one occasion a feline skin arrived at the collection of the Instituto Nacional de los Recursos Naturales Renovables y del Ambiente (INDERENA) that looked like the feline he had observed in the Cali Zoo. In 2001, a few months after the unfortunate death of Hernández-Camacho, Dr. Ruiz-García and Esteban Payán visited the Mammalogy Museum of the Instituto Alexander von Humboldt at Villa de Leyva (Boyacá, Colombia) to study the museum’s collection of cat skulls and pelts. While going through the drawers containing the skins of ocelots and margays, they discovered a “strange” skin (museum ID number 5857) that was classified as a *Felis* sp. The skin was in perfect condition and had been obtained from the Galeras volcano (altitude: 3100 m) in the Nariño department in southern Colombia in 1989. With the help of Dr. Yaneth Muñoz (curator of that collection), we obtained a small integumentary sample of the Nariño cat (3 cm^2^). We extracted DNA from this skin (2001 sample), and we began to preliminarily analyze 18 DNA microsatellites of this specimen and others from *L. pardalis*, *L. wiedii*, *L. tigrinus*, *L. geoffroyi*, *L. jacobita*, and the domestic cat. For *L. colocola* and *L. guigna*, we utilized microsatellite data from the literature. The Nariño cat was not clearly related to any of these species. Ruiz-García et al. [22] reported these preliminary data and the incomplete classification of the specimen. However, based on the principle of parsimony, the taxa with less differentiated alleles with respect to the Nariño cat were the Pampas cat and Geoffroyi’s cat. Furthermore, we hypothesized that it could be an undescribed form of Pampas cat, because this species is found as far north-central Ecuador, while Geoffroyi’s cat lives thousands of km away. Many years later, our prediction of the existence of the Pampas cat in Colombia was confirmed [23].

We continued to investigate the classification of this specimen and sent photographs of the skin to Dr. García-Perea, a renowned specialist in the Pampas cat [9]. Dr. García-Perea replied “This skin does not belong to any of the described forms of the Pampas cat. It must be one of those very rare forms of tigrina that appear on certain rare occasions”.

In 2007, we sent a small fragment of the skin that we sampled in 2001 to Daniel Cossíos at the University of Montreal for testing and analysis. At the time, Cossíos was a doctoral student working in Dr. Bernard Angers’s lab and was already collaborating with our research team. He extracted DNA and sequenced several mt genes (2007 sample). These first mt analyses did not clearly associate the Nariño cat to any other species of the *Leopardus* genus (the most related species for mt*D-loop* was *L. guigna*), although the mt*ND5* gene was the sister taxon to the Pampas cat. Some years later, Myreya Pinedo, a doctoral student of Dr. Ruiz-García, analyzed the mtDNA of the Nariño cat (the sample collected in 2001) when she was analyzing tigrina samples. Their results were practically identical to those obtained by Daniel Cossíos. We reported these results and announced the presence of a “strange” specimen [7]. To explore the specimen further, in 2017, we returned to the Instituto Alexander von Humboldt to obtain another integumentary sample (later known as the 2017 sample) of the Nariño cat. Our objective was to study and analyze nuclear genes in order to confirm the findings reached through the analysis of the mitochondrial genome. In 2023, an article was published specifically dedicated to the Nariño cat [8]. All of the published results came from the DNA obtained from the 2001 sample, because the 2017 sample results had not yet been fully obtained at that time. Some months later, another work appeared with some criticism relative to the Nariño cat [23]. Based on studying three small fragments of mtDNA, the authors claimed that the Nariño cat was simply a “traditional” northern tigrina. Again, we returned to the Instituto Alexander von Humboldt to obtain a new integumentary sample of the Nariño cat (October 2023; 2023 sample). Both the 2017 and 2023 samples were analyzed by a graduate student (Javier Vega) as part of his MSc research project. Some results of this analysis are presented in the results section of the current paper. More recently, another work from the same authors that previously criticized the finding of the Nariño cat [23] appeared with alleged new arguments against the differentiation of the Nariño cat from the traditional northern Andean tigrina [24].

Since 2018, we have increased the total number of samples gathered and analyzed of the Andean tigrina. For example, Ruiz-García et al. [7,8] analyzed the mt*ATP8* ± mt*16S rRNA* genes of 41 tigrinas (39 sampled in Costa Rica, Venezuela, Colombia, Ecuador, Peru, Bolivia, northwestern and northeastern Argentina and southeastern Brazil, along with two sequences obtained from GenBank). Additionally, the authors analyzed the mt*ND5* gene of 30 tigrinas (17 analyzed directly by the authors and 13 sequences obtained from GenBank) and the complete mitogenome of 18 tigrinas (17 analyzed by the authors and one obtained from GenBank). Within the tigrina term, we included *L. tigrinus*, *L. guttulus*, and *L. t. emiliae*.

In the current work, we present the analysis of the mt*ND5* gene of 63 tigrina specimens (37 northern Andean and Central American tigrinas, 4 temporal replicates from the Nariño cat, 13 *L. guttulus*, and 9 *L. t. emiliae*). Additionally, we present the analysis of the complete mitogenome of 42 “traditional” tigrina specimens (34 northern Andean and Central American tigrinas, 3 temporal replicates from the Nariño cat, 3 *L. guttulus*, and 2 *L. t. emiliae*). Thus, we have made a great effort to find more samples of northern tigrinas, even though it is a rare feline in the Andes, and we have basically duplicated the number of samples of tigrinas analyzed for the mt*ND5* gene and for the mitogenomes analyzed by Ruiz-García et al. [7,8].

The main aims of the present work are as follows: (1) To clarify whether or not the temporal samples obtained of the Nariño cat (four samples of the mt*ND5* gene and three of the mitogenomes) show the same molecular and phylogenetic results, and, if they do not, to provide an explanation as to why they are different (2) To present evidence (refuting claims by Astorquiza et al. [23], Lescroart et al. [19], and Marín-Puerta [24] by studying very few specimens) that the Andean tigrinas are made up of multiple polyphyletic lineages.

## 2. Material and Methods

### 2.1. Samples

Over the last 25 years, we have obtained and analyzed multiple types of tissue samples (integument, hair, teeth, bones, blood, and muscle) of the *Leopardus* genus throughout the Neotropics. For the current study, we analyzed two datasets.

The first dataset consisted of sequences of the mt*ND5* gene of 164 specimens of the genus *Leopardus*: (1) We directly sequenced 44 tigrina samples (35 northern Andean and Central American tigrinas, four temporal replicates from the Nariño cat [2001, 2007, 2017, 2023 samples], and five *L. guttulus*), 34 Pampas cat samples, *L. colocola* (these sequences represent many of the taxa reported for this species, *L. braccatus*, *L. budini*, *L. colocola*, *L. cruscinus*, *L. garleppi*, *L. pajeros*, *L. steinbachi*, and *L. wolffshoni*), six Geoffroyi’s cats, *L. geoffroyii*, six Andean Mountain cats, *L. jacobita*, eight margays, *L. wieddii*, and 23 ocelots, *L. pardalis*. Thus, we directly sequenced 121 specimens of the genus *Leopardus*. (2) We have taken data from GenBank as follows: two sequences of northern *Andean tigrina* [19,23], eight sequences of *L. guttulus* [15,16], nine *L. t. emiliae* [16,18,19,25], five *L. colocola* [16,18,25], nine *L. geoffroyi* [15,25,26], five kodkods, *L. guigna* [18,25,27], four *L. jacobita* [18,28], and one *L. pardalis* [18]. Thus, a total of 43 sequences of different species of the *Leopardus* genus were obtained from GenBank.

The second dataset consisted of mitogenomes from 102 specimens of the *Leopardus* genus. (1) We directly sequenced 37 tigrinas (32 were northern Andean and Central American tigrinas, three were temporal replicates of the Nariño cat [2001, 2017, 2023 samples] and two *L. guttulus*), 18 ocelots, seven margays, eight Andean Mountain cats, 11 Pampas cats representing diverse taxa within this species (*braccatus*, *colocola*, *steinbachi*, *munoai*), and seven Geoffroy’s cats. Thus, we directly obtained the mitogenomes of 88 specimens of the *Leopardus* genus. (2) We have taken data from GenBank as follows: Two northern Andean tigrinas [19]), one *L. guttulus* and two *L. t. emiliae* [18,19], one ocelot [18], two Andean cats [18], two Pampas cats [18], two Geoffroy’s cats [18], and two kodkods [18]. Therefore, 14 mitogenomes were obtained from GenBank.

### 2.2. Mitochondrial DNA

We extracted and isolated DNA from hair, blood, and muscle samples using the QIAamp DNAMicroKit (Qiagen, Inc. Germantown, MD, USA), and from bone, teeth, and skin samples using the NX-48S Tissue DNA kit (Nextractor^R^ NX-48S; Genolution, Seoul, Republic of Korea) for the mitogenomic analysis (a total of 16,756 base pairs, bp, including a fragment of mt*ND5* gene of 315 bp in length). Amplification of large mitochondrial DNA fragments before sequencing was carried out by Long-Range PCR, which minimizes the chance of amplifying mitochondrial pseudogenes from the nuclear genome (numts) [29,30]. We used a Long-Range PCR Kit (Qiagen, Inc., Valencia, CA, USA), with a reaction volume of 25 mL and a reaction mix consisting of 2.5 mL of 10 Long-Range PCR Buffer, 500 M of each dNTP, 0.6 M of each primer, 1 unit of Long-Range PCR Enzyme, and 50–250 ng of template DNA. PCR reactions were carried out in a PerkinElmer Geneamp PCR System 9600 Thermocycler (PerkinElmer, Waltham, MA, USA) and a Bio-Rad ICycler (Hercules, CA, USA). Cycling conditions were as follows: 94 °C for 5 min, followed by 45 cycles of denaturing at 94 °C for 30 s, then primer annealing at 50–57 °C (depending on primer set) for 30 s, and an extension at 72 °C for 8 min, followed by 30 cycles of denaturing at 93 °C for 30 s, annealing at 45–52 °C (depending on primer set) for 30 s, and extension at 72 °C for 5 min, with a final extension at 72 °C for 8 min. Four sets of primers [7] were used to generate overlapping amplicons from 3680 to 5011 bp in length, enabling a quality test for genome circularity [29]. Both mtDNA strands were sequenced directly using BigDye Terminator v3.1 (Applied Biosystems, Inc., Foster City, CA, USA). Sequencing products were analyzed on an ABI 3730 DNA Analyzer system (Applied Biosystems, Inc., Foster City, CA, USA). Sequences were assembled and edited using Sequencher 4.7 software (Gene Codes, Corp., Ann Arbor, MI, USA). Overlapping regions were examined for irregularities such as frameshift mutations and premature stop codons. A lack of such irregularities indicates an absence of contaminating numt sequences. We did not observe any such irregularity. The individual alignments were concatenated using SequenceMatrix v1.7.6 software [31] to create a master alignment. Base frequencies, amount of missing data, and the length of the final mitogenomes were checked with a custom Python script (scikit-bio v0.5.6). We aligned the final mitogenomes with Clustal X version 2.0 [32], and MUSCLE alignment plugin in Geneious R7.1 [33]. Both mt*ND5* and mitogenome sequences did not show chimeric regions (especially mitogenomes) because they were submitted to sliding window analyses with negative results.

The sequences we obtained are deposited in GenBank. Here are the accession numbers for the different *Leopardus* species we analyzed: PQ579121-PQ579150; OR725153-OR725174; MG230196-MG230233; OR522799-OR522799; OR557277-OR557296; PQ164764-PQ164774; OL615026-OL615080; MG230238-MG230242; MN005818-MN005921; MG23042-MG23045; OR725175-OR725177; OR673479-OR673494; OR520786-OR520799; OR685121-OR685121; MG230236-MG230237.

### 2.3. Statistical Analyses

#### 2.3.1. Genetic Distances

We used the Kimura 2P genetic distance [34] to determine the percentage of genetic differences between the different temporal Nariño cat samples (for mt*ND5* gene: 2001, 2007, 2017, 2023; for mitogenomes: 2001, 2017, 2023) and some selected specimens of tigrina. Seven tigrina samples were selected that represented different groups of tigrina that had some differentiated characteristic, morphological or molecular traits (a tigrina from Circasia, Quindío department in Colombia IAVH 7331; two tigrinas sampled that were run over on the northern highway in the outskirts of Bogotá and in La Calera, respectively; the two tigrinas reported by Astorquiza et al. [23] and Lescroart et al. [19]; one tigrina sampled in Intag, Imbabura province in Ecuador; and two tigrinas sampled in Costa Rica, Central America). Some of these specimens are shown in Figure 1. The Kimura 2P genetic distance was also estimated between these temporal samples of the Nariño cat and diverse species of the *Leopardus* genus (different taxa of Pampas cats, Geoffroy’s cat and kodkod). We compared these genetic distances because values higher than 6–11% are typical for distinct species [35,36]. We utilized this genetic distance because it is a standard measurement for barcoding tasks [37,38].

To compare the Kimura 2P genetic distances among the different selected sequences, we employed a variance estimation method with 100 bootstrap replicates for generating standard deviations when, also, a unique sequence is used in the pairs of comparisons [38]. We also used the consensus sequence of the temporal samples of the Nariño cat in the comparison (using the Kimura 2P genetic distance) of the mt*ND5* gene and mitogenomes with other *Leopardus* taxa.

#### 2.3.2. Phylogenetic Trees

jModeltest v2.0 [39], Kakusan4 [40], and MEGA X 10.0.5 [41] software were applied to determine the best evolutionary mutation model for the analyzed sequences of mt*ND5*, for different mt gene partitions, and for complete mitogenomes. All three programs offered identical results. The Akaike information criterion (AIC) [42] was used to determine the best evolutionary model for the relationships among the studied taxa of *Leopardus*. The GTR ± G model (General Time Reversible model ± gamma distributed rate variation among sites) [43,44] was found to be the best nucleotide substitution model for the two datasets used (mt*ND5* gene and mitogenomes).

We constructed two mitochondrial phylogenies for both aligned mt*ND5* gene sequences and aligned mitogenomes with maximum likelihood (ML) and neighbor-joining (NJ) procedures (total of four trees). For the ML tree, we used RAxML v8.2.12 [45]. RAxML was run with the GTRGAMMA model of nucleotide evolution and default parameters, and 1000 bootstrap replicates to calculate node support. For the NJ tree we used the Kimura 2P distance and two different software to compare results: MEGA X 10.0.5 software and Python package scikit-bio v0.5.6.

We showed herein two kinds of phylogenetic trees (NJ and ML) because they showed the most divergent results obtained. However, we also constructed Maximum Parsimony (MP) and Bayesian trees, but they were intermediate between NJ and ML trees and, in fact, very similar to the trees shown. Therefore, for brevity, only NJ and ML trees are yielded.

#### 2.3.3. Spatial Monmonier’s Algorithm Analysis

A spatial analysis was carried out with Monmonier’s Algorithm Analysis (MAA) [46] with AIS software [47] for different combinations of tigrina sets to determine the exclusiveness, or not, of the different samples of Nariño cat. This geographical regionalization procedure is used to detect the locations of putative barriers to gene flow by iteratively identifying sets of contiguous, large genetic distances along connectivity networks [48,49,50]. A Delaunay triangulation was used to generate the connectivity network among sampling points [51,52]. A graphical representation of putative “barriers” inferred by the algorithm is superimposed over the connectivity network to detect rapid identification of important geographical features reflected by the genetic dataset. In this case, we used two strategies. First, we utilized the mt*ND5* gene dataset with the tigrinas, excluding the tigrinas introgressed/hybridized by ocelot and margay, and excluding *L. guttulus* and *L. t. emiliae*. Second, we utilized the entire mitogenome dataset for all the tigrinas analyzed (including *L. guttulus* and *L. t. emiliae*). In both cases, we detected eight main geographical barriers in these molecular datasets.

#### 2.3.4. BAPS Analyses

Additionally, for both datasets (mt*ND5* gene and mitogenomes), we conducted a Bayesian Analysis of Population Structure with the software BAPS v6.0 [53]. Calculations were performed with the number of k clusters varying from 1 to 20. Five replicates were carried out for each k value. With the BAPS procedure, we adopted five strategies to determine the relationships of the temporal samples of the Nariño cat with other tigrinas and other *Leopardus* taxa. For the first strategy we focused on the mt*ND5* gene and analyzed 63 tigrinas (all the Andean and Central America tigrinas, the four temporal samples of the Nariño cat, *L. guttulus*, and *L. t. emiliae*). In the second strategy, also focusing on the mt*ND5* gene, we analyzed 21 tigrinas (all the tigrinas included in the cluster of Tigrina G1 and the four temporal samples of the Nariño cat). The third strategy focused on the mitogenomes of 22 tigrinas (all the tigrinas within the cluster Tigrina G1, and all the “strange” tigrinas which constituted solitary branches in the phylogenetic trees as well as the three temporal samples of the Nariño cat but excluding the tigrinas introgressed/hybridized with ocelots and margays, *L. guttulus*, and *L. t. emiliae*). The fourth strategy focused on the mitogenomes of 18 tigrinas (all the tigrinas within cluster Tigrina G1, and the three temporal samples of the Nariño cat). The fifth and last strategy focused on the mitogenomes of 17 tigrinas (all the tigrinas within the Tigrina G2 as well as all the “strange” tigrinas which constituted solitary branches in the phylogenetic trees as well as the three temporal samples of the Nariño cat). These analyses should show how important it is to compare the nucleotide sequences of the Nariño cat with different groups of other tigrinas to determine its differentiation from these other tigrina taxa.

Both, the results of the MMA and BAPS procedures were georeferenced in ArcGIS Pro 3.2.2 [54] matching the vectors and coordinates in the figures with a point layer including each sample geographic position, respectively. From this, vector layers were generated to visualize spatial genetic analyses overlaid on physical geography.

## 3. Results

### 3.1. Genetic Distances Between the Nariño Cat and Other Tigrina Specimens and Species of the Genus Leopardus

Table 1A shows the Kimura 2P genetic distances regarding mt*ND5* between Nariño cat samples (2001, 2007, 2017, 2023) and other tigrina specimens. The 2001 and 2007 samples differed slightly (0.5 ± 0.5%). That is, the sequences of the Nariño cat for the same mt gene obtained in two different laboratories were not identical but the difference was very small. The 2001 sample was genetically distanced from several selected tigrina specimens by 4.4 ± 1.3% (two tigrinas sampled near Bogotá that were run over) to 7.5 ± 1.8% (two Central American tigrinas from Costa Rica). Regarding other *Leopardus* species, such as *L. colocola braccatus* (5.5 ± 1.5%), and *L. colocola steinbachi* (5.2 ± 1.4%), these genetic distances overlapped with those obtained for the selected tigrinas. In other words, some specimens classified as *L. tigrinus* were more related to the Nariño cat than to the Pampas cats, but other tigrina specimens were more distant to the Nariño cat than to the Pampas cats. Other well-recognized small species of *Leopardus*, such as *L. guigna* (7.5 ± 1.8%) and *L. geoffroyi* (7.5–9.2 ± 1.8–2.0%), were more differentiated from the Nariño cat than to the selected tigrinas as well as the Pampas cat. The 2007 Nariño cat sample presented similar results. For example, the genetic distances between this sample and selected tigrina specimens ranged from 3.7 ± 1.2% (the two tigrinas run over near Bogotá) to 6.7 ± 1.7% (the two Central American tigrinas). The genetic distances with reference to other taxa (the two Pampas cat subspecies, the kodkod, and Geoffroy’s cat) resembled the results of the 2001 sample. The Nariño cat presented smaller genetic distances with selected tigrinas. However, the genetic distances between the Nariño cat and some Pampas cat taxa were smaller than with some tigrina specimens. These distances contrasted with those of the kodkod and Geoffroyi’s cat which were more genetically differentiated.

Comparatively, the genetic distances described above for the 2001 and 2007 Nariño cat samples were dissimilar to those for 2017 and 2023. For example, genetic distances of the 2017 sample with other tigrinas ranged from 7.6 ± 1.8% (tigrinas runover near Bogotá) to 10.2 ± 2.2% (the Central America tigrinas). The upper range was the same for kodkod (10.2 ± 2.2%). However, the genetic distance regarding Geoffroy’s cat was similar (9.7 ± 2.1%) to the previous cases. Again, dissimilar to the genetic distances regarding the Nariño cat samples of 2001 and 2007, the genetic distances with the Pampas cats were considerably reduced (2.3 ± 0.9% with *L. colocola braccatus* and 2.0 ± 0.8% with *L. colocola steinbachi*). Therefore, the 2017 sample presented the Pampas cat as the taxon most related with the Nariño cat. Incredibly, the genetic distances involving the 2023 sample were very different from those of the 2017 sample. Some selected tigrina specimens yielded the lowest genetic distances with the Nariño cat (2.3 ± 0.9%, the two tigrinas run over near Bogotá), whilst 4.5 ± 1.3% and 4.9 ± 1.4% were the genetic distances for the two tigrinas sampled in Antioquia and Caldas departments (Colombia) sampled by Astorquiza et al. [23], respectively. The greatest genetic distances with the selected tigrinas for comparisons with the Nariño cat were with the Central America tigrinas (6.0 ± 1.6%). The genetic distances among the 2023 sample and the two taxa of pampas cat analyzed clearly increased, (6.0 ± 1.6% [for *L. colocola braccatus*] and 5.6 ± 1.5% [for *L. colocola steinbachi*]). These values are on par with the highest genetic distances between the 2023 sample and the tigrinas most differentiated from the Nariño cat. The genetic distances of the 2023 sample and the kodkod and Geoffroy’s cat were slightly lower with reference to the samples from 2001, 2007, and 2017.

On the other hand, the genetic distances among all the tigrinas selected for this analysis (Table 2) were considerably smaller than any of the genetic distances between the various Nariño cat samples and these tigrinas. For instance, the smallest genetic distances among these tigrinas were 0 ± 0% (between the two tigrinas run over near Bogotá), 0.3 ± 0.3% (between the two tigrinas sampled by Astorquiza et al. [23]), 0.6 ± 0.5% (between another tigrina, different from the Nariño cat but also from the Nariño department and a tigrina sampled in Intag, northern Ecuador), and 1 ± 0.6% (between a tigrina from Circasia, Colombian Quindío department and the tigrina sampled in the Nariño department, different from the Nariño cat, and also between this tigrina and the two tigrinas run over near Bogotá). The largest genetic distances among these selected tigrinas were between the tigrina from Circasia (and the two tigrinas run over near Bogotá) and the two Central America tigrinas (3.4 ± 1.1%), and between the tigrina from Intag and the tigrinas from Central America (3.7 ± 1.2%). In any case, regardless of the year of sampling of the skin of the Nariño cat, it always showed values of genetic distances much larger than those reported by Astorquiza et al. [23] between the sample that they took of the Nariño cat and the only two tigrinas that they comparatively analyzed (one specimen from Antioquia and another specimen from Caldas) for the mt*ND5*. In fact, the authors affirmed in their paper that the genetic distance between the Nariño cat and these two tigrinas was 0. However, as we discussed below, based on the paper’s supplementary Tables [23], the author’s claim is inaccurate.

The full mitogenome of the 2007 sample was not analyzed, rather just a few mt genes were sequenced by D. Cossíos in Canada. Genetic distances between the 2001 sample and selected tigrinas, the Pampas cat taxa, the kodkod, and Geoffroyi’s cat were all similar (Table 1B). The smallest genetic distance with some of the selected tigrinas were 5.0 ± 0.08% (with the two tigrinas run over near Bogotá) and the largest genetic distances were 6.2 ± 0.1% with the majority of the selected tigrinas. The genetic distances with the two Pampas cat taxa ranged from 6.1 ± 0.09% to 6.4 ± 0.1%. These values completely overlapped with the greatest genetic distances for the selected tigrinas. Identically, the genetic distances with the kodkod (6.4 ± 0.1%) and Geoffroyi’s cat (6.2 ± 0.1%) completely overlapped with the genetic distances of the Pampas cats and the greatest genetic distances for some selected tigrinas. The 2017 sample showed a decrease in genetic distances with reference to the selected tigrinas. The smallest genetic distances were 2.7 ± 0.06% (with the two tigrinas run over near Bogotá) and the largest genetic distances were with a tigrina from Circasia (5.7 ± 0.09%) and another from Intag (5.8 ± 0.09%). The 2017 sample also showed a noteworthy decrease in the genetic distances with reference to the Pampas cat taxa (4.1 ± 0.08% for both taxa used). Thus, such as it occurred with the mt*ND5* for some years, the 2017 Nariño cat sample presented the smallest genetic distances with some determined tigrina specimens, but additionally, its genetic distance with the Pampas cat was smaller than with other tigrina specimens. The genetic distances with the kodkod and Geoffroyi’s cat were also slightly smaller (5.9 ± 0.09% and 5.6 ± 0.09%, respectively). Finally, the 2023 sample showed a stronger decrease in the genetic distances with reference to the tigrinas (the lowest genetic distances were 1.1 ± 0.04% with the two tigrinas that were run over near Bogotá, and the greatest genetic distances were 3.9 ± 0.07%, with the Circasia specimen, and 4.0 ± 0.07% with the Intag specimen). Also, the genetic distances with the kodkod and Geoffroyi’s cat were smaller (4.9 ± 0.08%, and 5.1 ± 0.09%, respectively). In contrast, the genetic distances regarding the Pampas cat taxa considerably increased in reference to the 2017 sample (5.9 ± 0.09% for both *braccatus* and *steinbachi*).

Table 2 shows the genetic distances among all the selected tigrinas for mitogenomes. The smallest genetic distances ranged from 0% (the two tigrinas run over near Bogotá), 0.22 ± 0.02% (the two tigrinas sampled by Astorquiza et al. [23]), to 0.9 ± 0.03% (the two tigrinas run over near Bogotá vs. the two tigrinas sampled by Astorquiza et al. [23]). The 2001 and 2017 Nariño cat samples always presented genetic distances with all the other selected tigrina that were larger than these values. However, the 2023 sample showed some genetic distances that were slightly higher than these values (1.1% with the two tigrinas run over near Bogotá). Based on this information, the 2023 sample is closely related to this Colombian tigrina nucleus, whereas the 2017 sample and, especially, the 2001 sample were comparably more distanced from this tigrina group. This analysis also supports that two of the selected tigrinas were differentiated from this Colombian tigrina nucleus (the Circasia and the Intag specimens). The Circasia specimen showed genetic distances ranging from 2.7 ± 0.06% (with the two tigrinas run over near Bogotá) to 4.0 ± 0.07% (with the two Central American tigrinas), whilst the Intag specimen showed genetic distances that ranged from 2.5 ± 0.06% (with the tigrina sampled in Nariño, but different from the Nariño cat) to 3.9% (with the two Central American tigrinas). Thus, the mitogenome analyses revealed that inside the specimens classified “a priori” as tigrina, there was certain observable genetic heterogeneity, which is not only caused by the Nariño cat.

In theory, the different samples taken in different years from the same skin of the Nariño cat should always offer the same results. But why does this not happen? In the discussion we offer an explanation of these inconsistent results.

Another interesting fact can be seen in Table 3. If we take the consensus sequence for the four samples of the Nariño cat obtained in 2001, 2007, 2017, 2023 for mt*ND5* and we compare it with different *Leopardus* taxa, the results show that the two smallest genetic distances were those with the two Pampas cat taxa (2.6 ± 0.8% with *L. c. braccatus*, and 2.9 ± 0.9% with *L. c. steinbachi*), whereas the genetic distance with the main tigrina nucleus was 3.2 ± 1.0%. It is remarkably that this consensus sequence for the Nariño cat at the mt*ND5* was also 3.2 ± 0.9% with some exemplars of Geoffroyi’s cats apparently introgressed with mtDNA of the Pampas cat that we sampled in Bolivia and Argentina. The genetic distances with the kodkod and with Geoffroyi’s cat were clearly higher (6.1 ± 1.4%, and 6.2 ± 1.3%, respectively). Nevertheless, this picture is different if we analyze the mitogenomic data. If we take the consensus sequence for the 2001, 2017, and 2023 years, the smallest genetic distance was with the main tigrina nucleus (Tigrina G1) (1.3 ± 0.05%), followed by the group of Geoffroyi’s cats introgressed with Pampas cat mtDNA (1.8 ± 0.04%). The genetic distances with the Pampas cats (4.2 ± 0.06%, 4.1 ± 0.05%) and with the kodkod and Geoffroyi’s cat (4.2 ± 0.08%, 3.8 ± 0.07%) were higher.

### 3.2. The Situation of the Different Nariño Cat Samples in the Phylogenetic Tree of the Leopardus Genus

We show four phylogenetic trees (Figure 2) of the diverse samples analyzed of the Nariño cat. The first is an ML tree based off mt*ND5* gene data (Figure 2A). As the remaining trees (both for mt*ND5* and complete mitogenomes), this tree shows that the specimens classified “a priori” as *L. tigrinus* did not form a monophyletic group (*L. guttulus* is not consider within the tigrinas in these trees). There were two main (and independent) groups of tigrinas. The large one, named Tigrina G1, contained two sub-groups, one composed of specimens from the Eastern Colombian Andean cordillera (as well as in some analyses by one Peruvian specimen) and another composed of specimens sampled in the Central and Western Colombian Andean cordilleras (including the two specimens analyzed by Astorquiza et al. [23], from Caldas and Antioquia departments). This last sub-group also contained some Ecuadorian specimens as well as the two Central American tigrinas from Costa Rica. Therefore, most specimens that “a priori” belonged to the previous defined taxa as *L. t. pardinoides* and *L. t. oncilla* were within Tigrina G1. A second tigrina group, named Tigrina G2, contained different specimens with a tigrina-like phenotype but molecularly very differentiated from those in Tigrina G1. All the specimens of this group were coming from Colombia and Ecuador. Curiously, a Geoffroy’s cat from the La Paz department in Bolivia also showed mtDNA of this group. This taxon should receive a new scientific name, but we will discuss this topic elsewhere. Additionally, we found several tigrina specimens enclosed in another well-defined *Leopardus* species. Within *L. pardalis*, there were tigrina specimens clearly introgressed in the past with ocelot mtDNA. However, it is very likely that young ocelots inhabiting high altitudes (ocelots with some morphological traits converging to those from the high-altitude tigrina) should be confounded with tigrinas. Some researchers, including those that provided samples to the authors of the current work, can mistake these ocelots for the “true” tigrina. The same occurred in the clade of *L. wieddii*. There were some clear phenotypically tigrina-like specimens introgressed historically with mtDNA of margay and others which should be margays living in high altitudes with convergent morphological characteristics of tigrina (this was the case of two specimens classified as *L. tigrinus* in Bolivia [55]) with typical mtDNA of its species. All the tigrina introgressed by mtDNA of ocelot and margay were specimens from Colombia as well as one specimen from Venezuela. The sequences of the tigrina taxa classified as *L. t. emiliae* from northeastern Brazil and obtained from GenBank were within the monophyletic clade of *L. colocola*. The fact that *L. t. emiliae* was introgressed by the Pampas cat was previously recognized [16,18,19] and it is not new. However, there were some other tigrina specimens that constituted solitary branches in the phylogenetic trees. For example, in the ML tree derived from mt*ND5* data (Figure 1A), there were the cases of tigrina 33 (Chingaza NP, Cundinamarca department, Colombia), tigrina 58 (Medio Sanjuan, Chocó department, Colombia), tigrina 66 from an unknown place in western Ecuador, and the different temporal samples of the Nariño cat. The 2001 and 2007 Nariño cat samples (both very similar) formed a solitary branch clearly differentiated from the clade formed by (*L. geoffroyi* ± *L. guigna*) ± (*L. guttulus* ± Tigrina G1). The 2023 Nariño cat sample also formed a solitary branch although less differentiated from *L. geoffroyi*, *L. guigna*, *L. guttulus* and Tigrina G1. Finally, the 2017 Nariño cat sample was inside the clade of *L. colocola* and *L. t. emiliae*.

The NJ tree derived from mt*ND5* data (Figure 2B) yielded similar results to the previous tree in reference to the tigrina specimens. The most relevant differences were the appearance of another tigrina specimen which formed a solitary branch (tigrina 64 from western Ecuador), the positional change of the solitary branches of tigrina 33, 58, and 66, and the positional change of three of the temporal samples from the Nariño cat. The 2001, 2007, and 2023 Nariño cat samples formed a solitary clade which is the sister clade of that formed by *L. colocola*, *L. t. emiliae*, and a little group of *L. geoffroyi* introgressed by Pampas cat mtDNA and the 2017 Nariño cat sample.

Thus, the phylogenetic trees based off the mt*ND5* gene data showed that three samples of the same skin of the Nariño cat taken in 2001, 2007, and 2023 were not associated with the two tigrina samples analyzed by Astorquiza et al. [23], as these authors claimed. They were part of the group Tigrina G1. The differentiation of the 2017 Nariño cat sample was even greater because it was clustered within the clade that integrated *L. colocola* ± *L. t. emiliae*.

The ML tree derived from the complete mitogenome data (Figure 2C) showed some traits identical to the previous *ND5* trees: the existence of two well differentiated tigrina taxa (G1 and G2) and some tigrina specimens forming solitary branches (although their positions in this tree could be different than in the previous cases), being the cases of tigrinas 33, 58, and 66. However, with the mitogenomes, other tigrina specimens appeared as differentiated from all the other *Leopardus* taxa and formed solitary branches. These were the cases of tigrina 20 (Circasia, Quindio, Colombia) and tigrina 42 (Intag, Ecuador), both related to Tigrina G1 but differentiated from the strict nucleus of Tigrina G1, and especially the tigrina specimen (tigrina 08) sampled in the Nevado del Ruiz (Caldas department, Colombia). That sample was collected at 4700 m and seemed to be introgressed with mtDNA from an ancestor of the clade of the ocelot and margay. The 2017 Nariño cat sample formed a group together with Geoffroy’s cat previously detected with the mt*ND5* gene that was introgressed with Pampas cat DNA. This was the sister cluster of the clade composed of *L. geoffroyi* and *L. guigna*. The 2001 Nariño cat sample (the original one) was the sister taxon of the previous clade integrated by (*L. geoffroyi* + *L. guigna*) + (introgressed *L. geoffroyi* + 2017 Nariño cat sample). In contrast, to these two Nariño cat samples as well as that observed with the mt*ND5* gene, the 2023 Nariño cat sample experienced a noteworthy closeness to the clade formed by Tigrina G1.

The NJ tree based off complete mitogenome data (Figure 2D) was like the previous tree. The two most important differences were the position of Tigrina G2 (the most differentiated taxa within *Leopardus* in all the other trees) and in this tree is the sister taxa of the clade comprising ocelots, margays, Andean cat, and Pampas cat. The other difference is the position of two tigrina specimens (tigrinas 20 and 33). The positions of the Nariño cat samples were the same as in the previous tree: the 2017 sample, together with introgressed Geoffroyi’s cats, formed the sister taxa of *L. geoffroyi* + *L. guigna*. The 2001 sample was the sister taxa of this clade. Again, the 2023 sample integrated a branch slightly differentiated from the strict nucleus of Tigrina G1.

All four Nariño cat samples analyzed for the mt*ND5* were noteworthily differentiated from Tigrina G1. (where the two tigrinas analyzed by Astorquiza et al. [23] are enclosed), Based on the analysis of the complete mitogenomes, two samples (2001, 2017) were also significantly differentiated from Tigrina G1, but the 2023 sample was considerably closer to Tigrina G1 (although not identical as incorrectly claimed by Astorquiza et al. [23]).

### 3.3. The Differentiation of the Nariño Cat Using the Monmonier Algorithm (MA) and BAPS

The MA is an interesting procedure to detect geographical barriers which split the continuity of genetic pools. When the MA was applied to the mt*ND5* data (excluding the tigrinas introgressed with margay and ocelot, and *L. guttulus*, and *L. t. emiliae*), we determined eight potential geographical barriers for the tigrinas (Figure 3A). The first barrier extended from Costa Rica, crossing the Antioquia department (Colombia) up to the northern Boyacá department (Colombia). The second barrier isolated the tigrina specimen sampled in Junín (Peru). The third barrier comprised an area centered on the Nariño department (Colombia) with an extension to the east, reaching the Caquetá department. This area also showed an extension to the north, reaching the Valle del Cauca department (Colombia). The fourth barrier isolated an area extending from Costa Rica to the Pacific area of Colombia and Ecuador. The fifth barrier delineated an area including the departments of Caldas and Quindío (Colombia) extending to the northern Valle del Cauca department (Colombia). The sixth barrier isolated an Ecuadorian area which enclosed the provinces of Santo Domingo de Tsáchilas and Pichincha. The seventh barrier delineated an area in the Colombian departments of Huila and northern Valle del Cauca. An eighth barrier created an area compressed by the Colombian departments of Caldas, Cundinamarca and Huila. Therefore, we observed an important number of potential barriers in the area where these tigrinas were sampled. The complex geology of the northern Andes is the main cause of these potential geographical barriers. One of these barriers delineated the area where the Nariño cat was sampled (the third barrier).

When we carried out the MA procedure for the overall mitogenomes (in this case including *L. guttulus* and *L. t. emiliae*) (Figure 3B), the picture was as follows. The first barrier delineated an area extending from the extreme eastern Colombia and covering basically all northern and north-eastern Brazil (Ceara state, etc.). Clearly, this area corresponds to the geographical distribution of *L. t. emiliae*. This tigrina taxon was introgressed with the mtDNA from the Pampas cat and was the tigrina taxon with the highest degree of genetic differentiation in reference to the other tigrinas. The second barrier delineated a small area in the Colombian departments of Huila and Caquetá. The third barrier created an area which extends from northern Ecuador (Imbabura province), across the Nariño department, up to the Valle del Cauca department (both in Colombia). The fourth barrier delineated an area extending from the northern Valle del Cauca department, and includes the Colombian departments of Risaralda, Chocó, Antioquia, until Costa Rica. The fifth barrier basically follows the outline of the distribution area of *L. guttulus* (states of Santa Catarina, Paraná, in southern Brazil). The sixth barrier created an area integrated by western Ecuadorian provinces (Cotopaxi, Santo Domingo de Tsáchilas, Pichincha) that extends north to Costa Rica, and south to the Peruvian department of Cajamarca. The seventh barrier overlaps the second, third, and six barriers including again the Colombian departments of Huila and Nariño and extends to central Ecuador. The eighth barrier delineates an area that is essentially central Peru (Junín department). Therefore, two of the barriers involve the Nariño department where the Nariño cat was sampled. In fact, one of these barriers (barrier 3) was more important than barrier 5, which is what separated *L. guttulus* (now considered a full species) from the other tigrina taxa. That is, for both mt*ND5* and mitogenome datasets, some important geographical barriers were detected with the MA in the Nariño department (Colombia), precisely in the area where the Nariño cat was sampled.

With BAPS analysis, we adopted five different analytical strategies. In the first strategy, we considered only the mt*ND5* gene with 63 specimens (including all 37 northern tigrinas we analyzed irrespective of their positions in the phylogenetic trees, the four temporal samples of the Nariño cat, nine *L. t. emiliae*, and 13 *L. guttulus*). There were seven optimal partitioned groups [log(ml) = −2162.972]. These groups were as follows. (1) One group contained all the Tigrina G1 including three temporal Nariño cat samples (2001, 2007, 2023) as well as the two tigrinas from Astorquiza et al. 2023 [23]. (2) A second group contained the entire Tigrina G2. (3) A third group included all the tigrina introgressed/hybridized with ocelot or margay mtDNA together with some tigrina specimens that were in a unique branch in the phylogenetic trees (i.e., tigrinas 58 and 64). (4) A fourth group contained all the *L. t. emiliae* plus the 2017 Nariño cat sample. (5) A fifth group contained all the *L. guttulus*’s specimens from Brazil. (6) A sixth group had two rare tigrina specimens (tigrina 33 from the Cundinamarca department in Colombia, and tigrina 66 from an unknown origin in western Ecuador). (7) A seventh contained three *L. guttulus* that we sampled in Argentina (Figure 4A). Therefore, in this analysis, three of the four temporal Nariño cat samples belonged to the most expanded group (Tigrina G1 in Central America, Colombia, and Ecuador). However, the 2017 Nariño cat sample was associated with the *L. t. emiliae* cluster which was introgressed by the Pampas cat. It is important to note the remarkably strong differentiation between the Tigrina G1 and Tigrina G2 groups in the northwestern Andes, the strong differentiation of tigrinas 33 and 66 in Colombia and Ecuador from the other tigrinas, and the genetic differentiation between the Brazilian and Argentinian *L. guttulus*.

In the second strategy, also with the mt*ND5*, we used 21 tigrinas (17 tigrinas plus the four temporal Nariño cat samples) belonging “a priori” to Tigrina G1. There were three optimal partitioned groups [log(ml) = −463.359]. The three groups were as follows. (1) One group contained all the specimens showing as G1 in the phylogenetic trees, including the two tigrinas from Astorquiza et al. (2023) [23]. (2) The second group contained fourth temporal Nariño cat samples, irrespective of the sample year. (3) A third group consisted of tigrina 33 from the Cundinamarca department in Colombia (Figure 4B). In contrast to the previous analysis, although the molecular marker was the same, the four temporal Nariño cat samples were perfectly differentiated from Tigrina G1. Again, tigrina 33 was differentiated from other Colombian and Ecuadorian tigrinas.

The third strategy considered the complete mitogenomes with 22 tigrinas, including those previously observed in the phylogenetic trees as Tigrina G1 and those placed in a unique branch of the phylogenetic trees (tigrinas introgressed/hybridized with ocelot and margays as well as Pampas cats, *L. t. emiliae*, and *L. guttulus* were not included). There were three optimal partitioned groups [log(ml) = −12,931.882]. (1) One group contained Tigrina G1 along with the three temporal Nariño cat samples (2001, 2017, 2023). (2) A second group only contained tigrina 33. (3) The third group consisted of tigrina 08 (from Nevado del Ruiz in the Caldas department in Colombia), and tigrina 66 from some unknown location in western Ecuador (Figure 4C). Thus, this analysis did not differentiate the three temporal Nariño cat samples from Tigrina G1.

Nevertheless, the fourth strategy, which focused on the complete mitogenomes of 18 tigrina (Tigrina G1 and the three temporal Nariño cat samples [the other “strange” tigrinas were excluded]), the situation was clearly different. There were two optimal partitioned groups [log(ml) = −6390.086]. (1) One group contained all the specimens from Tigrina G1 together with the 2023 Nariño cat sample. (2) The second group consisted of the 2001 and 2007 temporal Nariño cat samples (Figure 4D). In this case, the original Nariño cat samples were significantly differentiated from Tigrina G1.

The fifth strategy considered the complete mitogenomes of 17 tigrinas (Tigrina G2 and all the rare tigrinas in a unique branch in the phylogenetic trees, and the three temporal Nariño cat samples). There were two optimal and significant partitioned groups [log(ml) = −18,088.748]. (1) One group contained all the Tigrina G2 specimens. (2) The second group had the three temporal Nariño cat samples together with all the other “strange” tigrinas that were in different parts of the *Leopardus* phylogenetic tree (tigrinas 08, 20, 33, 42, 58, and 66) (Figure 4E). This analysis clearly shows that Tigrina G2 is highly differentiated from the “strange” tigrinas which did not belong to Tigrina G1.

Henceforth, these five BAPS analyses put forward that the position of the Nariño cat is clearly dependent on the groups of taxa and species of the *Leopardus* genus to which it is compared. The comparison by Astorquiza et al. [23] of the Nariño cat to only two tigrinas and two Pampas cats was insufficient.

## 4. Discussion

### 4.1. Why Is the Phylogeny of the Leopardus Species Complicated?

The first interesting question on this topic concerns the recent radiation of the current felines in general, and the *Leopardus* genus in particular. Modern felines have a relatively recent common ancestor (11 million years ago, mya) that rapidly gave rise to distinct clades, with a very rapid process of speciation within them [56,57]. In the case of the *Leopardus* (the most speciose genus within the Felidae [18]), its radiation was during the Pliocene. Johnson et al. [56] estimated the beginning of the diversification of this genus to have occurred around 2.91 mya (4.25–2.02 mya). Li et al. [18] estimated this time to be around 3.14 mya (5.03–1.64 mya), whereas Ruiz-García et al. [7] estimated a value of 3.91–3.06 mya. More recently, Lescroart et al. [19] estimated the basal split of the *Leopardus* genus to have occurred around 4.45 mya (5.33–3.60 mya). All these estimations were highly similar and showed that the current species of *Leopardus* genus have recently evolved. This rapid process of speciation for the species of *Leopardus*, as well as their particular spatial patterns and their demographic dynamics, seems to have been jointly driven by climate fluctuations and habitat specialization (extreme changes in temperature, humidity, hydrology, geology, and biomes were present in the Neotropics during the end of the Pliocene and the Pleistocene [58,59] with different ecological adaptations leading to distinct radiations. But this relatively recent and rapid speciation process entails several problems that have a very important effect on the attempt to reconstruct the phylogeny of this genus. The first is a phenomenon named incomplete lineage sorting (ILS). Different recent species may share a good deal of the same genetic variability (e.g., same alleles or haplotypes). Therefore, ancestral shared genetic variation could be common under a scenario of multiple speciation events that proceed in quick succession, as occurred within *Leopardus*. In turn, ILS is typically invoked as the default cause of phylogenomic incongruence found in many organisms [60]. In the *Leopardus* genus, ILS has been repeatedly detected [18,19,57]. But the genealogical discordance largely attributable to ILS should be augmented by two other events which occurred in *Leopardus*. They are the extreme situations of the same phenomenon. One is the existence of a punctual historical ancestral introgression (PHAI). Introgression has been previously demonstrated among different species of felines [61], including nuclear versus mitochondrial genealogical discordance in *Leopardus* species [16,18,57]. In the introduction, we refer to the case of Trigo et al. [16], which provided strong evidence of PHAI between Pampas cat and populations of tigrina in northeastern Brazil (*L. t. emiliae*). A typical pattern of PHAI is that the mtDNA of one species appears in another species (in the case of Trigo et al. [16], the crossing of a *colocola* female with a tigrina male) and the offspring repetitively only interbreeding with the lineage of one of the parents (in the commented case, the subsequent crosses occurred only within tigrina). Over time, only evidence of maternal mtDNA from another species will remain, but no traces of introgression will remain in the nuclear genome. Furthermore, the phenotype will typically be one of the parental species (see, for instance, the case of the southern Central America coati, *Nasua narica*, in Panama and northern Colombia, introgressed with mtDNA from *Nasuella* (=*Nasua*) *olivacea* [62,63,64]). The other extreme is repetitive introgression or recent hybridization (RI-RH). Even though hybridization began a long time ago, it would have continued until recent times in such a way that mtDNA of another species and nuclear introgression would be detected. Morphology of these recent introgressed or hybridized specimens could be intermediates between parents. A conspicuous example is the one reported by Slon et al. [65] of a fossil of a young Denisovian girl (Denisova 11, Denny), with mtDNA from a Neanderthal and a nuclear genome, half from the father (Denisovian), and half from the mother (Neanderthal). The example of Trigo et al. [15,16,17] with relatively recent and current hybridization between *L. guttulus* and *L. geoffroyi* could be like this case. The analysis by Li et al. [57] revealed very high fractions of topological discordance along the genome of four *Leopardus* species, more so than in any other felid lineage due to ILS, PHAI, and RI-RH.

In fact, Lescroart et al. [19] detected high gene flow between Geoffroy’s cat and *L. guttulus* as well as historical introgression between ocelot and the ancestral lineage which gave place to Geoffroy’s cat, the kodkod and the majority of tigrinas. In the authors’ first test they estimated the admixture fraction between Geoffroy’s cat and *L. guttulus* to be 18%. They detected another strong signal (6%) of admixture between ocelot and the ancestral lineage. With a second test, they confirmed a high fraction of gene flow between Geoffroy’s cat and *L. guttulus* (40%). Shared ancestry between ocelot and the group of Geoffroy’s cat, kodkod and tigrinas, as well as Pampas cat, was estimated at 13%.

In our case, we detected several interesting cases of PHAI, and RI-RH implying some tigrinas with the ocelot-margay lineage. For example, we found a tigrina-like specimen from Apia (Risaralda department, Colombia, altitude: 1800 m). The mitogenome of this specimen was mixed with the mitogenomes of 19 ocelots. It should be a case of RH or, this specimen is an ocelot but with convergent morphology with the Andean tigrinas living at high altitudes. Two other tigrina-like specimens (one from Mesa Bolivar, Merida Sate, Venezuela, altitude of 1100 m; and the other from Pitalito, Huila department, Colombia, altitude of 1400 m) formed the sister cluster to the ocelots. These specimens could represent an intermedium case of PHAI and RI-RH. The same occurred with two other tigrina-like specimens (one from La Virginia, Risaralda, Colombia, altitude of 1100 m; and the other from San Agustin, Huila department, Colombia, altitude of 2000 m). They were the sister cluster of the margays. Two other tigrina-like specimens (one from Caucasia, Antioquia department, Colombia, altitude of 300 m, and the other from Valdivia, Antioquia department, Colombia, altitude of 1350 m) should represent RH between tigrina and margay or they were margays with convergent tigrina phenotypes. We analyzed two specimens classified as tigrinas in Bolivia (Florida and Puerto Limón, Santa Cruz, altitudes of 2000 and 1500 m, respectively [55]), but they were margays with some tigrina convergent coat characteristics. Finally, we discovered two clear cases of PHAI. One specimen from Nevado del Ruiz (Caldas department, Colombia, altitude of 4700 m) and the other specimen from Medio Sanjuan (Chocó Department, Colombia, altitude of 120 m). That is, two specimens from very different habitats with full characteristics of tigrinas but with mtDNA capture from an ancestor of the ocelot-margay lineage.

Margay and ocelot are sister species for mtDNA [18,56]. Thus, it is highly probable that both species share mt haplotypes by ILS and/or PHAI events that occurred post speciation. Ocelots are considerably larger than tigrinas and probably depredated them. Therefore, there is a low probability of a breed based on a tigrina male with an ocelot female. There is a higher probability of a breed between a tigrina male with a margay female (more similar in size). These margay females shared ocelot haplotypes by ILS and/or previous PHAI-RI-RH between ocelots and margays.

Another interesting finding in this research were the cases of four *L. geoffroyi* (two from the Bolivian departments of La Paz and Beni, and two from the Argentinian La Pampa province in Lihue Calel National Park) with differentiated Pampas cat mtDNA. Their morphologies were of a Geoffroy’s cat. It could be a clear case of PHAI.

The extent of these phylogenetic discordances explains the difficulties in reconstructing the phylogeny within *Leopardus*. Given the rapid diversification and, consequently, the short internal branches in the phylogeny of *Leopardus*, it is likely that large portions of the genome are subject to ILS [59] as well as PHAI, and RI-RH, which have likely added to the complexity of the evolution of the *Leopardus* genus. The tendency for PHAI and RI-RH in the *Leopardus* species is facilitated by the strong karyotype similarity among these felines coupled with a very recent genetic divergence [66,67,68]. One special consequence, especially of PHAI and RI-RH, is that it can lead to the introgression of genetic material from one lineage into another. In the context of rapid and continuous speciation, PHAI and RI-RH may be prevalent because of incomplete reproductive isolation between emerging lineages [69] as well as the influence of novel genetic material, which can potentially boost the evolvability of the recipient taxon [70].

### 4.2. The Phylogenetic Situation of the Nariño Cat, of Other Tigrinas, and Why the mtDNA Sequences of the Nariño Cat Were Not Consistent Throughout the Years

We played with two variables to try to resolve this enigma: the year in which the samples from the skin of the Nariño cat were obtained, and the molecular marker used.

In all the analyses carried out using the 2001 and 2007 samples for mt*ND5*, the results were basically the same. The 2001 DNA sample was processed in Bogotá at a time that we did not analyze samples of *L. colocola*, *L. geoffroyi*, *L. guigna* or *L. jacobita* in our lab. Therefore, the Nariño cat skin could not to be contaminated with DNA of these species. We were analyzing DNA samples from ocelot, margay and domestic cats but there was no evidence of contamination from these species in the Nariño cat’s DNA. In fact, when we discovered this skin in the Mammalogy Museum of the Instituto Alexander von Humboldt at Villa de Leyva (Colombia), it was in the same drawer where the skins of ocelots and margays were deposited. However, in no case was DNA contamination from these species observed. Some years later (in 2007), we sent a small integumentary sample (from the sample we obtained in 2001) of this specimen to D. Cossíos, who, at the time, was a doctoral student whose research focused on *L. colocola*. We were in collaboration to conduct molecular research on this species. Potentially, in Canada, the 2007 sample could have been contaminated with Pampas cat’s DNA, but it is important to note that DNA extracted in Canada offered similar results to what we obtained in 2001 with no evidence of DNA contamination from the Pampas cat.

Both DNAs for mt*ND5* showed equidistant genetic distances regarding the tigrinas (3.7–6.7%) and the Pampas cat (5.2–5.5%). We selected mt*ND5* because it is the mitochondrial gene with the most specimens and sequences of *Leopardus* reported in GenBank and therefore provided the opportunity for the greatest number of comparisons with our sequences.

With complete mitogenomes, we only used the 2001 sample because D. Cossíos did not obtain the mitogenome of the Nariño cat from the small skin sample that we sent him in 2007. The results showed significant genetic distance differences regarding both tigrina (5.0–6.2%) and Pampas cat (6.1–6.4%), although the genetic differentiation in reference to the second species was slightly greater.

In 2017, surprisingly, the results obtained at the mt*ND5* gene showed a significant relationship increase with the Pampas cat (2.0–2.3%) and a conspicuous differentiation from the heterogeneous set of tigrinas (7.6–10.2%). During the processing of the 2017 Nariño cat sample in our lab in Bogotá, we were analyzing samples of *L. colocola*, *L. geoffroyi*, *L. guigna*, etc., and therefore we cannot exclude the possibility of contamination with Pampas cat DNA. With complete mitogenomes, this differentiation was less conspicuous, and the genetic distances were relatively equidistant for the tigrinas (2.7–5.8%) and for the Pampas cat (4.1%). Thus, the possible contamination with Pampas cat DNA would seem to affect the results in reference to the individual mt*ND5* gene, but less so with the complete mitogenomes.

Even more remarkable were the results from a new sample of the Nariño cat that we obtained in October 2023. In June of that same year, we had published our findings regarding the discovery of the Nariño cat [8]. In response to this publication, some Colombian biologists, through social networks, and through certain newspapers (with journalists eager for prominence), initiated and carried out an intense public smear campaign against the discovery of this new taxon. However, science does not solve its enigmas by initiating public smear campaigns but instead by publishing results and hypotheses that can be falsified (or ratified repeatedly) in peer reviewed scientific journals. For this reason, we never publicly responded to those criticisms, and we resampled the Nariño cat skin to repeat the investigation. As aforementioned, the results with the 2023 sample were more contradictory. In this case, at the mt*ND5* gene, the results were similar to those of the 2001–2007 samples. The genetic distances with the tigrinas (2.3–6.0%) and the Pampas cats (5.6–6.0%) indicate a slightly greater resemblance towards the tigrinas than did the 2001–2007 samples. However, the mitogenomes results were heavily skewed towards the tigrinas (1.1–4.0%) and more differentiated toward the Pampas cat (5.9%). Regardless, both genetic sets (for all the Nariño cat samples) support that the genetic relationships between the Nariño cat and the other tigrinas analyzed, as well as among all the tigrinas analyzed, were extremely diverse and heterogenous. This fact did not occur with the other species of the *Leopardus* genus, with some small exceptions with *L. geoffroyi* (investigated elsewhere).

Additionally, if we used the consensus sequences for the mt*ND5* gene (2001, 2007, 2017, 2023 samples) and the consensus mitogenome (2001, 2017, 2023) for the Nariño cat regarding the other tigrinas and the Pampas cats, the results were divergent. For the mt*ND5* gene (small sequences but with many specimens compared), the Nariño cat was more related to the Pampas cat (2.6–2.9%) than to the tigrinas (3.2%), whilst, with mitogenomes (large sequences but with a lower specimen number), the Nariño cat presented a stronger relationship with the tigrinas (1.3%) than with the Pampas cat (4.2%).

Obviously, these different results obtained for the Nariño cat (2001–2007, 2017, 2023) have important repercussions in its systematic relationship with other species or taxa of the *Leopardus* genus. The NJ tree with the mt*ND5* gene showed 2001, 2007, and 2023 Nariño cat samples as the sister taxon of the Pampas cat (a unique species, not many species as suggested by Nascimento et al. [6] [this will be discussed elsewhere]). The 2017 Nariño cat sample was placed inside *L. colocola*. The ML tree with the same gene showed the samples from 2001, 2007, and 2023 as a transition group between the Pampas cat and the cluster integrated by *L. geoffroyi*, *L. guigna*, *L. guttulus*, and Tigrina G1. The 2017 Nariño cat sample was again placed inside *L. colocola*.

In reference to Tigrina G1, here is a brief commentary. Within this cluster, we detected two sub-clusters, one composed of specimens mainly sampled at the Eastern Colombian Andean cordillera and another composed of specimens sampled in the Central and Western Colombian Andean cordilleras as well as in the Ecuadorian Andean cordilleras. The first sub-cluster should be named *Leopardus tigrinus pardinoides* or *Leopardus pardinoides pardinoides* following de Oliveira et al. [71], because the holotype of *pardinoides* is thought to be from the Eastern Andean Colombian cordillera in Bogota [72]. The second sub-cluster should be *Leopardus tigrinus oncilla* or *Leopardus pardinoides oncilla* following de Oliveira [71] (the holotype with type locality: Volcán de Irazu, Costa Rica [73]). Within this sub-cluster were the two Central American tigrinas as well as the two Colombian tigrinas analyzed by Astorquiza et al. [23] and Lescroart et al. [19] and other tigrinas from Colombia (Western and Central cordilleras) and Ecuador. With this new result, the proposal of Marín-Puerta [24] that the two Colombian Andean tigrina analyzed by Astorquiza et al. [23] (which do not come from the Eastern Andean Colombian cordillera) could be classified as *Leopardus pardinoides pardinoides* is unfeasible. Both, Central American and the Andean tigrina analyzed by Astorquiza et al. [23] and Marín-Puerta [24] would be classified as *L. pardinoides oncilla* and the scientific name of *Leopardus pardinoides pardinoides* would only be intended for the tigrinas from the Eastern Andean Colombian cordillera. If our 2001–2007 samples of the Nariño cat represented the most real situation, then, the Nariño cat not belonging to *L. pardinoides.* Contrarily, if the 2023 sample of the Nariño cat will turn out to be the closest to reality, then, the Nariño cat would be a third subspecies within *L. pardinoides*, *Leopardus pardinoides narinensis*.

Additionally, a new nomenclature (scientific name or binomen) is required for Tigrina G2. This group of tigrinas is highly differentiated from Tigrina G1 and is dispersed in different areas of Colombia and Ecuador (even a specimen of *L. geoffroyi* from La Paz department in Bolivia was carrying this mtDNA), although its morphology is not clearly differentiated from Tigrina G1. Another possibility is that Tigrina G2 may be the traditional *L. tigrinus tigrinus* from Venezuela, Guyana, and northern Brazil that has penetrated the Andes and that does not have such a marked morphological differentiation of the *pardinoides* group according to Nascimento and Feijó [5].

The NJ and ML trees for mitogenomes offered the same results regarding the samples of the Nariño cat. The 2001 sample was the sister taxon of the clade composed of *L. geoffroyi* and *L. guigna*; the 2017 sample was inside a cluster of *L. geoffroyi* introgressed by *L. colocola*, and the 2023 sample was the sister taxon of the cluster of Tigrina G1.

We suggest the following to explain this systematic confusion. As aforementioned, the 2017 sample (that showed the major relationship with *L. colocola*) was analyzed together with many samples of Pampas cat and specimens of *L. geoffroyi* introgressed with Pampas cat DNA. It is possible that contamination from the Pampas cat affected the DNA of the 2017 sample. However, based on data and reasoning, we think that the 2023 sample was contaminated with DNA from Tigrina G1. This determination is based on the following information. (1) In 2001 and in 2017, the skin of the Nariño cat was in perfect condition (Figure 5). Although 16 years passed between samplings, the skin was intact and had the same appearance. However, when we returned to sample the skin in 2023 (six years later), its appearance was noticeable deteriorated (Figure 5). The texture of the fur had changed and its intense reddish colors and the black hood that covered its head and back were noticeably diluted. We know that after 2017, other research groups intensively manipulated the skin, in addition to taking samples from it, which opens up the possibility that it was contaminated with Tigrina G1 mtDNA, especially if these research groups were conducting molecular analyses with tigrinas. (2) In 2023, the skin was placed with other skins of tigrinas, some of them obtained in more recent years than when we sampled it the first time in 2001. This makes the contamination with DNA from other tigrina skins a distinct possibility. The contamination with Pampas cat DNA at the Instituto von Humboldt is nigh impossible because there are no skins or materials from the Pampas cat at the institute.

In contrast to the 2017 and 2023 samples, which were likely contaminated either with Pampas cat DNA or with Tigrina G1 DNA, we strongly suggest that the DNA samples obtained independently in our lab in Bogota (2001) and in the Canadian lab quoted previously (2007) and which had very similar results, were not contaminated with the DNA of other cat species. Which means, the 2001 and 2007 samples were those that offered the most realistic results, confirming the Nariño cat as a “rare” lineage of *Leopardus*. In fact, contamination is a plausible hypothesis, but direct experimental validation is required (i.e., replication in independent labs). Although in all the cases this was not done, the analysis of the samples 2001 and 2007 at the mt*DNA* in Bogotá (2001) and in Canada (2007) was a case of replication in independent labs because the DNAs obtained in both labs were from the same piece of skin, although the sequences were obtained in different years. Thus, we believe that DNA contamination should explain as a main hypothesis the inconsistences of the Nariño cat in the diverse phylogenies obtained.

In short, the samples taken from the single specimen that allowed the Nariño cat to be defined across different years (2001, 2007, 2017, 2023 for the mt*ND5* gene, and 2001, 2017, 2023 for mitogenomes) to see if the genetic results of this specimen were consistent in diverse phylogenetic trees revealed inconsistent results since the phylogenetic relationships of this cat with other *Leopardus* taxa were variable. The results obtained with the 2001 and 2007 samples (DNA obtained and independently analyzed in two different labs in Colombia and Canada from the same piece of skin) always showed the same pattern for the mt*ND5* gene. Depending on phylogenetic trees obtained with different procedures, the Nariño cat was either the sister group of [(*L. geoffroyi* + *L. guigna*) + (*L. guttulus* + Tigrina G1)] or the sister group of *L. colocola*. For the mt*ND5* gene, the 2017 sample was always within the *L. colocola* independently of the phylogenetic procedure employed, meanwhile the 2023 sample behaved like the 2001 and 2007 samples. For the mitogenomes, the 2001 sample presented the Nariño cat as the sister group of *L. geoffroyi* + *L. guigna*. Contrarily, the 2017 sample was related to a group of *L. geoffroyi* introgressed with *L. colocola* DNA (two possible explanations: these specimens of *L. geoffroyi* showed mtDNA from a very old process of introgression with *L. colocola*-or ancestors of *L. colocola*- and therefore these sequences were not identical to the current mt sequences of *L. colocola*; or these *L. geoffroyi* sequences were also partially contaminated with *L. colocola* DNA but in a lower degree than the 2017 sample for the mt*ND5* gene), meanwhile the 2023 sample was always the sister group of the Tigrina G1 group independently of the phylogenetic procedure employed. These inconsistent results could be explained by the fact that the 2017 sample could have been heavily contaminated with *L. colocola* DNA for both mt*ND5* gene and mitogenomes. On the other hand, the 2023 sample showed no contamination by DNA from other *Leopardus* taxa but presented possible contamination with Tigrina G1 DNA but to a lesser extent than the *L. colocola* DNA contamination of the 2017 sample. As the sequencing of mt*ND5* and mitogenomes were independently carried out, this shows that the degree of contamination for both kinds of molecular markers and for both samples (2017 and 2023) were probably also of different magnitude. This leads us to conclude that the sequences obtained in 2001 and 2007 are the most realistic and, therefore, the Nariño cat could be a new taxon within the *Leopardus* genus.

### 4.3. Potential Geographical Barriers Affecting the Tigrinas in the Northern Andean Cordilleras

The MA analyses showed, for both mt*ND5* and mitogenomes, that in the northern Andean cordilleras there are many potential barriers for the dispersion of small cats, many more than, for instance, in Eastern and Southern Brazil, where in recent decades new species of tigrinas have been confirmed [5,15,16]. For instance, the area where the Nariño cat was sampled corresponds to the North-Andes Biogeographic province, in an endemism highland zone named “Nudo de los Pastos” characterized by a cold, or very cold, climate with a volcanic geomorphology origin [74]. But in the Andean cordilleras of Colombia, Ecuador, Peru and Bolivia, there are evidence of possible Pleistocene refuges, geological deflections, or places of abrupt climate change, which may have had some influence on the diversification of the tigrinas. Traditionally, in Colombia, several important Pleistocene refuges have been considered [75,76,77,78], some of them in the Andes or in the foothills of the Andes, extending to the Ecuador: the Chocó refuge (Pacific coast in Colombia and Ecuador; tigrina 58 comes from that area in Colombia and tigrinas 64 and 66 come from that area in Ecuador), Nechí-San Lucas refuge (Antioquia, Córdoba, Bolívar), Sierra Nevada de Santa Marta refuge (Magdalena, Guajira), Catatumbo refuge (Norte de Santander, Venezuela), Sarare refuge (Arauca, Boyacá, Norte de Santander), Imerí-Vaupés refuge (Guainía, Vaupés, Venezuela and Brazil), and Napo refuge (Colombian, Ecuadorian and Peruvian Amazon). Nevertheless, many other dry or wet Pleistocene refugia of smaller size have subsequently been determined in Colombia [79,80]. Regarding dry refuges and those related to the Andes, we have Santa Fé de Antioquia (Cauca River Canyon), Alto Valle del Cauca (Cauca), Patía and Juanambú Canyon (Cauca and Nariño; the Nariño cat is geographically related to this refuge), Cúcuta and Alto Valle del Táchira (Norte de Santander and Venezuela), Chicamocha Canyon (Norte de Santander), Alto Valle del Magdalena (Huila and Tolima, and Tatacoa Desert), and Dagua Canyon (Valle del Cauca). Regarding Colombian wet refuges not cited above, we have Macuira refuge (NE Guajira), Villavicencio refuge (Boyacá, Cundinamarca, Meta and Casanare. Tigrina 33 comes from that area), Florencia refuge (Caquetá. One specimen of Tigrina G2 comes from that area), Putumayo refuge included in the Napo refuge by Haffer [75], Kofan refuge (it should be a southern part of the Napo refuge in the Colombo-Ecuadorian frontier, reaching Nariño), Mitu refuge (Vaupés), Apaporis refuge (Vaupés), Huitoto refuge (Caquetá, Putumayo), Sinú-San Jorge refuge (Antioquia, Córdoba), Huila refuge (Huila), and Tacarcuna-Aspavé-El Limón (Serranía del Darién, Colombian-Panamanian frontier). The existence of differentiated forms of tigrinas as the Nariño cat and others of the “rare” tigrinas detected in this study in the Andean area (for instance Tigrina G2 and tigrinas 08, 33, 58, 64 and 66) come from several of the refuges commented. Other mammal species, including felines like *L. colocola* and *L. jacobita* [10,11,81], have shown as some Andean barriers differentiated their populations. For instance, the Huancabamba or Cajamarca Deflections (in northern Peru) or the Abancay Deflection (in central-southern Peru) have had some influences in reptiles and mammals [82,83]. The Andean area between 18° and 23° S corresponds to an extremely arid belt that separates a northern Andean area, with summer rains, from a southern Andean area, with winter rains [84]. This zone is a barrier between mammalian populations or species. These are the cases of the vicuna, *Vicugna vicugna* [85], the lesser grison, *Galictis cuja* [86], the long-tailed weasel, *Mustela frenata* [87], and the pampas cat, *L. colocola* [10,11]. The case of the Andean cat, *L. jacobita* [11,81], is extremely interesting to see the phylogeographic effect that different areas of the Andes have on the genetic characteristics of a feline. The Andean cat separated from the common ancestor of other *Leopardus* cats at 1.8 MYA [56]. However, the common ancestor of their current mt haplotypes dated at approximately 217,000 to 290,000 years ago. This period covers an inter-glacial stage known as the marine oxygen isotope stage 7 (MIS-7, 0.19 to 0.24 MYA) and the glacial MIS-8 (0.24 to 0.3 MYA) [88]. The current lack of mtDNA variation in the Andean cat could be the result of numerous lineage extinctions due to demographic events that took place during MIS-7 or MIS-8. During interglacial periods, the habitat available for organisms specialized in high altitudes is expected to decrease (i.e., *Canis simensis*) [89], suggesting that MIS-7 is the most likely period for the loss of these lineages in the Andean cat. Similar processes could have occurred with the Andean tigrina that could have helped the appearance of multiple molecular lineages although morphologically they are all similar (the same happens with the Andean cat). Other authors [10,90,91,92] have proposed other Pleistocene refuges as the Titicaca refuge (southern Peru and Bolivia), the Chuquisaca-Jujuy refuge (Bolivia and Argentina) and the Mendoza-Neuquén refuge (Argentina, Chile). Samples of tigrinas from these areas are extremely limited, but perhaps when more tigrinas from these areas are analyzed, the Pleistocene refuges could be relevant to finding more genetic divergence within the Andean tigrina.

The BAPS analyses also revealed an interesting trend. The differentiation of the Nariño cat from other groups depends on which species or taxa were included in these analyses. With the mt*ND5* gene, with all the tigrinas analyzed irrespective of their positions in the phylogenetic trees, as well as with *L. emiliae* and *L. guttulus*, the 2001, 2007, and 2023 Nariño cat samples were indifferentiable from Tigrina G1. Contrarily, if we only analyzed the specimens of Tigrina G1 and the four samples of the Nariño cat, all four samples of the Nariño cat behaved as a distinct group from Tigrina G1. With the mitogenomes, including all the tigrinas from Tigrina G1, the “rare” tigrinas, and the three temporal samples of the Nariño cat, the last behaved as a tigrina from Tigrina G1. However, if the “rare” tigrinas were eliminated from the analysis, the 2001 and 2017 Nariño cat samples formed a different group from Tigrina G1 (2024 sample followed behaving as a tigrina from Tigrina G1). Henceforth, it is very important to compare the Nariño cat, as well as all the “rare” tigrinas, with the maximum number possible of specimens of all the existent taxa of the *Leopardus* genus to infer its real systematic position.

### 4.4. The Nariño Cat and Its Scientific Name

The systematic of tigrinas has always been very confusing. Schreber [93,94] first used the scientific name *Felis tigrina* and illustrated this species with a plate named “Le Margay” [95]. It was based on an exemplar from Cayenne, French Guiana. Thus, biologists have long been confused about what a tigrina is and what a margay is. Gray [96] described a supposed new species, *Felis pardinoides*, with “India” as its type locality, but later changed the type locality to Bogota, Colombia [72]. In the same period, *Felis guttula* was described, and southern Brazil was listed as its type locality [97]. Additional tigrina taxa were described in the beginning of the 20th century, including *F. p. oncilla* (Costa Rica), *F. p. andina* (Ecuador), *F. carrikeri* (Costa Rica), *F. p. emerita* (Venezuela), and *F. emiliae* (type locality: Ipu, Ceara State, Brazil). Allen [98] used the genus *Margay* and defined two additional holotypes of tigrina: *M. t. elenae* (Antioquia department, Colombia) and *M. caucensis* (Cauca department, Colombia). Later, the taxonomic confusion increased between tigrinas and margays. Two genera were defined: *Margay* and *Oncilla* [99]. Margay had two species: *Margay tigrina*, with three subspecies (*Margay tigrina tigrina* (the original *F. tigrina*; geographical distribution: eastern Venezuela and northern Brazil; today *L. tigrinus*), *Margay tigrina wiedii* (southern Brazil, southwestern Colombia; today *L. wiedii*), and *Margay tigrina vigens* (Para, Brazil; today *L. wiedii*) and *Margay glaucula* with two subspecies: *Margay glaucula glaucula* (Mexico; today *L. wiedii*) and *Margay glaucula nicaraguae* (Nicaragua; today *L. wiedii*). *Oncilla* had three species: *Oncilla pardinoides* with five subspecies: *O. p. pardinoides*, *O. p. oncilla*, *O p. andina*, *O. p. emerita*, and *O. p. elenae*. *O. p. pardinoides* inhabits Bogota, Colombia to southeastern Brazil (until recently referred to as *L. tigrinus*, although some specimens classified by other authors as *Felis geoffroyi* and *Felis guigna* were classified in this taxon; [99]). The second subspecies, *O. p. oncilla*, inhabits Costa Rica. (*O. p. oncilla* contains the two holotypes of *F. p. oncilla* and *F. carrikeri* from Costa Rica; [99]; until recently it was referred to as *L. tigrinus*). The third subspecies, *O. p. andina*, was found in Ecuador (until recently referred to as *L. tigrinus*). The fourth subspecies, *O. p. emerita*, is distributed in Merida, Venezuela and southward in the Eastern Colombian Andes to the Huila Department (until recently referred to as *L. tigrinus*). The fifth subspecies, *O. p. elenae*, inhabits Antioquia, Colombia (until recently referred to as *L. tigrinus*). The second *Oncilla* species was *Oncilla caucensis* (Cauca, Colombia; until recently referred to as *L. tigrinus*). The third *Oncilla* species was *Oncilla guttula*, with two subspecies: *O. g. guttula* (southern Brazil; today referred to as *L. guttulus*) and O. g. *emiliae* (Ceara, northeastern Brazil; today referred as *L. t. tigrinus* or *L. emiliae*). The systematic confusion surrounding the tigrina only increased in the following decades. Cabrera and Yepes [100] considered one unique species of tigrina that was classified in the genus *Noctifelis*. They identified it as *Noctifelis pardinoides*. However, others recognized a different species (*Noctifelis guigna*) in this genus [99]. Weigel [101] classified 10 subspecies of margay (currently known as *L. wiedii*) within *F. tigrinus*. For instance, the subspecies of margay from Para, Brazil to Guyana was named *L. t. tigrinus*, meanwhile the “true” tigrina was classified within in *Oncifelis pardinoides*, which included four subspecies: *O. pardinoides oncilla* (Costa Rica), *O. pardinoides pardinoides* (western Venezuela, Colombia, Ecuador, and Peru), *O. pardinoides emiliae* (Ceara, Brazil), and *O. pardinoides guttula* (southern Brazil). Finally, the tigrina classification we used until recently was adopted with *L. tigrinus* with four subspecies, as the introduction mentions [14,15,16]. Also recently, as was explained in the introduction, Trigo et al. [15,16,17] genetically differentiated *L. guttulus* from *L. tigrinus*.

Pelage characteristics and morphometrics from a large collection of *L. tigrinus* [5] support the existence of three morphogroups of tigrina. Morphogroup I contains specimens from Central America, as well as specimens from northern, northwestern, and western South America (Costa Rica, Colombia, Venezuela, Ecuador, Peru, Guyana, Suriname, and northwestern Argentina). The color of these tigrina is brown, and ranges from orangish brown to yellowish-brown. These individuals have a grayish brown under layer, a white or light gray venter, and medium-sized rosettes that form medium-sized oblique bands. The bands are arranged in a scapular-inguinal direction on the sides of the body. Morphogroup II contains specimens from the northeastern and central Brazilian area. Their color ranges from a light yellowish brown to pale yellow or pale grayish buff. They also have small-sized rosettes that rarely form oblique bands. The rosettes have thin and discontinuous black rims. Morphogroup III contains individuals from southern Brazil, Paraguay and northeastern Argentina. Overall, they have a dark yellowish-brown ground color, which is lighter on the sides of the body. They also have a white or light gray venter, and small rosettes on the sides of the body. The northeastern Brazilian tigrina population (morphogroup II) could be considered a new *Leopardus* species, *L. emiliae* [5], although Lescroart et al. [19] considered this taxon still a subspecies of *L. tigrinus*. Further, morphotype III refers to *L. guttulus*, whereas morphotype I would correspond to *L. tigrinus*. Therefore, for this study [5], *L. t. oncilla* and *L. t. pardinoides* are junior names absorbed by *L. tigrinus*. Recently (2024), another work added information on the possible number of tigrinas [71]. These authors performed an analysis integrating ecological modeling, biogeography, and phenotype of the four traditional recognized subspecies [14,15,16] and presented a new multidimensional niche depiction of the species and concluded that the Central American tigrina and the Andean tigrina conformed a unique species, *L. pardinoides* Contrarily to that claimed by Nascimento and Feijó [5], this last work [71] did not recognize *L. t. emiliae*, which is absorbed by *L. tigrinus*.

In this complex systematic situation of the tigrinas appeared the Nariño cat to make it even more complicated. Marín-Puerta [24] criticized that the use of the binomen of *L. narinensis* was first used in 2018, but at that time he considered that this scientific name was not available according to the rules of the International Code of Zoological Nomenclature (ICZN), due to the no assignation of a type specimen and the lack of a detailed description (however, the same author commented in his work, contradictorily, that the first time that the binomen *L. narinensis* was mentioned was in 2020 by Pinedo-Castro and Ruiz-García [102]) and that *L. narinensis* became available in 2023, when the description of the species was published in accordance with ICZN rules. This is a half-truth due to the ignorance that this author and his professors have of how the events occurred with the binomen of the Nariño cat. Ruiz-García et al. [8] explicitly commented in the Data Availability Statement that the datasets generated and analyzed in that study were available from the corresponding author on reasonable request at his e-mails. Therefore, we have always had the best aptitude to collaborate and inform our colleagues regarding the Nariño cat. Another issue is that these colleagues have never approached us to find out how this finding came about and, therefore, they will begin to speculate or imagine things that have nothing to do with reality. The first attempt to publish the manuscript where the Nariño cat was explicitly defined was in PLOS One. When the first two reviewers gave their opinion (which were positive) and the respective changes in the original version of the manuscript were incorporated, the academic editor of the journal explicitly told the authors that they had to obtain the Life Science Identifiers (LSID) from Zoobank (the online registration system for the ICZN) because without this requirement the article could not be published. For this reason and complying with all the rules requested by ICZN (type specimen and its correspondent description), we uploaded all this information to Zoobank. However, finally, the negative comments of a third reviewer arrived and the manuscript finally was not accepted by the aforementioned journal, but the LSID was obtained with the rules requested by ICZN at that moment. This was probably not the best chronology of events, but it is how it happened. Also, Marín-Puerta [24] criticized Ruíz-García et al. [8] because supposedly described *L. narinensis* based on a single skin collected in the Galeras volcano at Nariño (Colombia) and the description included only limited external data and the mitochondrial DNA information was not used. This is again half-truth. External morphological data are limited, but only one skin of this possible new taxon was recovered, but the non-use of molecular data does not correspond to reality. Textually, Ruíz-García et al. [8] said: “…Molecular diagnosis: Species-level diagnostic characters for the Nariño cat were observed in the two mitochondrial genes for which there are more sequences in neotropical cats (mt*ATP8* and mt*ND5* genes) and yielded the following seven synapomorphies: At the *ATP8* gene (three synapomorphies) in the nucleotide positions 8530 (T), 8594 (A), and 8597 (T). At the *ND5* gene (four synapomorphies) in nucleotide positions 12,506 (A), 12,715 (A), 12,737 (A), and 12,749 (A). These seven synapomorphies differentiated these two mt genes of the Nariño cat from those of all the other species of the genus *Leopardus* and from the domestic cat and the jaguarundi…”.

We have more examples where new mammalian taxa have been defined with very partial biological data (and without molecular data). One example, Schwartz [103] with only two specimens (a skeleton without skin and one skull) described a new species of primate (*Pseudopotto martini*), and new genera (*Pseudopotto*) and even a new family. The exact provenance of the two specimens was unknown. Schwartz [103] placed both specimens in a single species but commented that the two specimens might be two different species of this new genus. Obviously, some authors considered this finding as controversial. Stump [104] stated that some traits but not all of *Pseudopotto martini* were present in some pottos, mainly from western populations of *Perodicticus potto*. Obviously, these findings from very partial biological material will be always controversial as the Nariño cat case is. Originally, when we obtained the molecular data from the 2001 and 2007 samples, we only talked about the Nariño cat as a new molecular lineage that had not previously been detected in the *Leopardus* genus. Nonetheless, some colleagues and reviewers encouraged us to define this linage as a new species, using the case of *Pseudopotto martini* as example. But really, we agree with the Nobel laureate, Svante Pääbo [105] that when reticulated evolution is so important without a clear existence of reproductive barriers as potentially can occur in the case of the tigrinas (as also in the case of humans; recall that Pääbo never gave a binomen to the Denisovians or Neanderthals because they hybridized among themselves and with the current humans), it is better not to speak of species with fixed scientific names (Linnaeus conceived scientific names as binomen for organisms that were created and fixed and did not evolve) and we consider better to talk about lineages with common names or distinctive numbering. One of some classical examples could be that of the *Vandiemenella viatica* species group [106,107]. This wingless-matchstick morabinae grasshopper genus contains two nominal species (*V. viatica* and *V. pichirichi*) and 11 provisional taxa (*viatica*17, *viatica*19, P24X0, P24XY, P24XY-translocation, P25X0, P25XY, P45bX0, P45bXY, P45cX0, and P50X0). A similar option could be adopted for some of the new tigrina taxa we find in this study (i.e., Nariño cat, tigrinas 08, 58, 64, 66) until it can be determined exactly what kind of taxa they represent.

### 4.5. What Conclusions of Astorquiza et al. [23] and Marín-Puerta [24] Regarding the Nariño Cat Were Incorrect?

A few months after the paper by Ruiz-García et al. [8] was published, Astorquiza et al. [23] authored a paper on the Nariño cat. Likewise, more recently, Marín-Puerta [24] showed a BSc thesis directed by one of the authors of the previous article, where they used very similar arguments against the differentiation of the Nariño cat from other tigrinas. Nevertheless, there are inconsistences in their work, as discussed next.

(1)The authors only compared their DNA sequences of the Nariño cat with two Colombian tigrinas and with two Pampas cats [23], and with the same two Colombian tigrinas, and one Central American tigrina, one *L. guttulus*, one *L. t. emiliae*, and one *L. colocola* [24] (these four last sequences taken from the study of Lescroart et al. [19]). We have shown that the systematic relationships of the Nariño cat (and the other “rare” tigrinas) depends on the number of other specimens of the *Leopardus* genus used in the comparison. Clearly, the comparison with only two northern Andes tigrinas is not enough. It is relevant to mention the genomic-whole analysis used by Lescroart et al. [19] to try resolving the phylogeny of the *Leopardus* genus. Nonetheless, they only used two Colombian and one Central American tigrinas (three specimens) to carry out their analyses. Although they correctly discriminated against these northern tigrinas from *L. guttulus* and *L. t. emiliae* (as Marín-Puerta [24] did) they concluded that only a group of tigrina is present in northern-western South America and Central America. Herein we showed at the mt*ND5* gene, analyzing 164 specimens of *Leopardus*, which included 63 specimens representing the “traditional” tigrina, that the 37 northern Andean and Central American tigrinas were divided, at least, into six different groups (without counting the Nariño cat). The largest group (Tigrina G1) was represented by 16 specimens (the two Colombian tigrinas reported by Astorquiza et al. [23] and Lescroart et al. [19] and the Central American tigrinas were inside this group). The second group (Tigrina G2) was composed of nine tigrinas while the third group (eight specimens) was composed of tigrinas introgressed/hybridized with the margay and the ocelot. The other four tigrinas made up isolated branches. For mitogenomes, we analyzed 102 specimens of the *Leopardus* genus, including 42 specimens of tigrina. The 34 northern Andean and Central American tigrinas were divided, at least (without counting the Nariño cat), into eight different groups. The largest group (Tigrina G1) was represented by 13 specimens (including the tigrinas from Astorquiza et al. [23] and Lescroart et al. [19] and the Central American tigrinas). The second group (Tigrina G2) was composed of eight tigrinas. The third group was integrated by five tigrinas introgressed/hybridized by the margay and ocelot and later appeared as five isolated branches formed by one or two “rare” tigrina(s).

Clearly, the three northern Andean and Central American tigrinas used by Astorquiza et al. [23] and Lescroart et al. [19] represented the majority group, but they totally underestimated the genetic diversity of minority groups that went completely unnoticed in their sampling. Whenever the number of groups within a determined taxon are ignored, it is better to sample and analyze many specimens from many localities (covering the full geographic range of the taxon) for a reduced number of genetic markers than a few specimens from a reduced number of localities for complete metagenomes [64,108].

(2)Astorquiza et al. [23] commented that the three fragments of the mtDNA they studied (*16S rRNA*, *COX2-ATP8*, and *ND4-ND5*) totaled 845 bp. This contrasts with the 16,756 bp we studied for the mitogenome dataset. Of these three molecular markers they studied, the mt*ND5* gene was the best discriminator among species of the *Leopardus* genus. The primers that they employed for *ND4-ND5* only amplified 145 bp, whereas we obtained nearly 1800 bp (although we used 315 bp of this gene to carry out the comparisons with the sequences deposited in GenBank). Thus, their study of the Nariño cat not only used far fewer samples, but they also studied a much smaller portion of mitochondrial DNA compared to the study by Ruiz-García et al. [8]. In fact, some of the mitochondrial genes used by Astorquiza et al. [23] (such as *16S rRNA* and *ATP8*) did not highly differentiate the Nariño cat from Tigrina G1. However, other mt genes such as *ND5*, *ND6*, *Cyt-b*, or *COX3* differentiated well both small spotted cats. Marín-Puerta [24] used the same mt sequences generated by Astorquiza et al. [23]. Therefore, his molecular data lacks novelty and has exactly the same problems as those shown by Astorquiza et al. [23].(3)Astorquiza et al. [23] textually affirmed that the Nariño cat showed 100% identity with sequences of the two Colombian tigrinas that they studied, although its identity with *L. colocola* sequences ranged from 88 to 98%. An analysis of the authors’ Supplementary Tables demonstrates that their affirmation is incorrect. Furthermore, these supplementary Tables contain mistakes. For example, Supplementary Table S3, shows that the genetic distance for *COX2-ATP8* between the Nariño cat and the two tigrinas was 0. Nonetheless, the authors reported a genetic distance of 0.27% between the two Colombian tigrinas. If the sequences of the two tigrinas were not identical, then one of both tigrinas could not be identical to the sequence of the Nariño cat. Thus, it is materially impossible for the Nariño cat to be simultaneously identical to both tigrinas. Clearly, the authors made some mistakes in their numerical estimations. Second, Supplementary Table S4 for *ND4-ND5* showed that the Nariño cat was differentiated from one of the tigrinas by 0.47%. Supplementary Table S5 indicates that the Nariño cat was differentiated from one of the tigrinas by 0.16% and from the other by 0.08%. These genetic differentiation values were very small, but they do not support a 100% genetic identity between the Nariño cat and the two Colombian tigrinas as the authors claimed. Marín-Puerta [24] also repeated that the Nariño cat and the two Colombian tigrinas used by Astorquiza et al. [23] and by himself had a 100% of genetic identity. However, in his ML tree showed that the Nariño cat was in a different branch in reference to the two Colombian tigrinas. Thus, it is materially impossible that the Nariño cat had a 100% of genetic identity with these two Colombian tigrinas.(4)Another refutable affirmation of Astorquiza et al. [23] and Marín-Puerta [24] concerns the morphotype of the Nariño cat. They used the argument that the Nariño cat skin was classified by Nascimento and Feijó [5] within morphogroup I. Effectively, the Nariño cat by the disposition of its rosettes is more related to morphogroup I than to morphogroup II (*L. t. emiliae*) or to morphogroup III (*L. guttulus*). But this statement is misleading. Nascimento and Feijó [5] also showed, for instance (Figure 10, p. 246), that a skin of margay (NMNH388255, Rio Cunucunuma, Belen, Venezuela) was extremely like the skin of a tigrina from morphogroup I (USNM374861, El Manteco, Bolivar, Venezuela). Therefore, based on Nascimiento and Feijó [5], this margay skin could be classified in Morphogroup I, being a margay a different species from a tigrina. That is, *Leopardus* species other than tigrinas could also be classified in morphogroup I using the reasoning of Nascimento and Feijó [5]. The fact that the Nariño cat by the disposition and color of its rosettes belongs to morphogroup I does not mean it is not strongly different in other coat characteristics to other tigrinas from morphogroup I. Indeed, if anyone looks at Figure 1 from Astorquiza et al. [23], they can easily observe that the Nariño cat skin (Figure 1e) is clearly different from the Pampas cat skin (Figure 1a–c), but it is just as easy to observe that the Nariño cat skin is totally distinguishable from the other tigrina skin analyzed by these authors (Figure 1d). It is certain that the Nariño cat skin has a similar size to other tigrina skins, and no skull (or other bones) are associated with this skin to be compared with material of other tigrina and this limited quantitative morphological comparisons weaken the case for the Nariño cat’s distinctiveness. However, an indisputable fact is that the overall physical appearance of the skin of the Nariño cat is totally distinguishable from any other skin of the different taxa recognized within the *Leopardus* genus. On the other hand, Marín-Puerta [24] added an alleged new proof against the skin differentiation of the Nariño cat. He commented that the intense reddish coat color of the Nariño cat’s skin is an artifact of bad preservation (he forgets the texture of the coat, the intense black hood of the head and the middle area of the back, the non-existence of the white ventral fur of the traditional tigrilla, etc.) and that this skin was exposed to smoke or prolonged exposure to intense heat, considering the unusual coloration of the hindfeet bones. This is a very unbelievable argument for several reasons. First, the preservation of this skin was excellent until at least 2017 and it was not subjected to chemical or physical treatments that would have mistreated it because, if it had been, it would have been difficult for us to obtain DNA of such good quality as to obtain its complete mitogenome. Second, to say that it was smoked or that it was subjected to very high temperatures is totally incomprehensible since this skin comes from a very cold place in the Andean Páramo at 3100 masl and where the technique of smoking hunted animals is not usually used, unlike what happens in the Amazon. When an animal smokes, the blood is destroyed and the reddish color in the bones of the legs is a symptom that the blood was preserved because it was exposed to very low temperatures that preserved well its morphological characteristics as well as its DNA (i.e., from the Peruvian mummies found at high altitudes). In fact, other skins from other tigrina obtained in the IAVH museum and that were treated with chemical procedures or heat did not offer DNA of good quality or in high quantities as the Nariño cat skin did.(5)Astorquiza et al. [23] claimed that the Nariño cat sequences they generated would be deposited in GenBank upon submission of their paper. However, to date these sequences have not been submitted to GenBank and, therefore, it is impossible to compare them with our Nariño cat’s sequences. Likely, Marín-Puerta [24] claimed that the genetic data used in Ruíz-García et al. [8] has not been previously published and lacks traceability, making the study impossible to replicate. This statement is not entirely true. The senior author of the current article (MR-G) has uploaded to GenBank around 460 sequences of specimens of the *Leopardus* genus, whilst the senior author or director of [23,24] (HR-CH) is collaterally associated with seven sequences of specimens of the same genus. On the other hand, Marín-Puerta [24] has not either uploaded in GenBank the sequences of the Nariño cat that he worked in his thesis. Perhaps, they may not be able to replicate the present study because they do not have as large numbers of northern Andean tigrina samples and with geographic origins as diversified as what we show in this study.

In short, the systematic position of the Nariño cat is still opened to scientific debate. We strongly suggest that the original DNA extracted in 2001 and 2007 was not contaminated. Thus, based on our analysis of the sample analyzed in 2001 and 2007, the Nariño cat has a different mitochondrial lineage within the *Leopardus* genus. A negative condition is that at the present time, the skin may be highly contaminated with Tigrina G1 DNA due to the mismanagement of the skin in recent years. It would also be very important to be able to analyze its nuclear DNA to study its possible ILS, PHAI and/or RI-RH events with other taxa of the *Leopardus* genus. In fact, reliance only on mitochondrial data limits insights into nuclear introgression; integrating nuclear markers would strengthen claims. For this reason, Ruiz-García et al. [8] analyzed six DNA microsatellites of the Nariño cat, and they were related to *L. geoffroyi* and to *L. guigna*. However, we now know that the five *L. t. pardinoides* that were used for that DNA microsatellite analysis, were Colombian tigrinas introgressed/hybridized by margay and ocelot and, therefore, they were not representative of “pure” Andean tigrina lineages. Given our new findings and explanation of the limitations of previous studies (i.e., Astorquiza et al. [23] and Martín-Puerta [24]), it is important to analyze additional nuclear DNA of the Nariño cat to fully elucidate the Nariño cat’s real taxonomic position. If we consider that the 2001–2007 DNA was of good quality without any type of contamination, it is possible that the Nariño cat was basically a tigrina carrying ancient mtDNA from an extinct lineage of *Leopardus* (ILS or PHAI with a tigrina lineage from a species or lineage of *Leopardus* already extinct). On the other hand, if we assume that the highest quality DNA came from the most recent sample (2023), then the Nariño cat mtDNA differs little from other tigrinas (especially Tigrina G1, and the two tigrinas from Astorquiza et al. [23]), and the markedly different physical appearance of its coat is likely the result of recent hybridization (RH) with other feline species (nuclear gene sequences are required to confirm this). We recognized that this study stresses genetic data because this is the unique data that we have on this new lineage or species, together the skin which served to extract DNA. No bones, skulls, or other similar skins associated with other tissues are found until the present moment. Therefore, all the knowledge that we can infer about the Nariño cat is basically from a genetic point of view. However, we recognize that morphology, pelage patterns, craniometrics, and ecology remain decisive in felid species delimitation. Henceforth, the unique way to absolutely clarify if the Nariño cat, the Tigrina Group 2, and other “rare” specimens of tigrina (tigrinas 08, 33, 58, 64, 66, for example) are new species, hybrids or linages introgressed with mtDNA of extinct species of the *Leopardus* genus is with Whole Genome Sequencing (WGS). Nevertheless, the finding of new haplotypes or haplogroups previously not found in neither of the current recognized *Leopardus* species is a powerful indication (although not definitive proof) of the possible presence of a new lineage or species (see, for instance, the discovery of a new human species, the Denisovians, when fossils of these humans were still not discovered [65]).

Regardless of the case, our analysis shows the existence of multiple lineages within the presumed “a priori” northern Andean tigrina that represent a series of potential taxa that have gone unnoticed due to a lack of sampling of this polyphyletic Andean feline.

## 5. Conclusions

Ten conclusions can be drawn from this study: (1) The samples taken from the single specimen that allowed the Nariño cat to be defined across different years (2001, 2007, 2017, 2023 for the mt*ND5* gene, and 2001, 2017, 2023 for mitogenomes) to see if the genetic results of this specimen were consistent in diverse phylogenetic trees revealed inconsistent results since the phylogenetic relationships of this cat with other *Leopardus* taxa were variable. Probably, the samples from 2017 and 2023 were contaminated to some extent with DNA from other felines and this leads us to conclude that the sequences obtained in 2001 and 2007 are the most realistic and, therefore, the Nariño cat could be a new taxon within the *Leopardus* genus. (2) The *Leopardus* phylogeny is complex due to the existence of repetitive ILS, PHAI and RI-RH events related with weak reproductive barrier mechanisms between many species of the *Leopardus* genus and because the *Leopardus* diversification is recent. We detected two tigrinas with clear cases of PHAI and RI-RH with ocelots and two with clear cases of PHAI and RI-RH with margays, although the first two events could have reached the tigrinas via margays due to ILS events of the latter with ocelots. We also detected two tigrinas with clear cases of PHAI with some ancestor of the ocelot-margay lineage. (3) For the mt*ND5* gene, the 37 northern Andean and Central American tigrinas we analyzed were divided, at least, into six different groups (without counting the Nariño cat). For mitogenomes, the 34 northern Andean and Central American tigrinas analyzed were divided, at least (without counting the Nariño cat), into eight different groups. Henceforth, there is strong evidence for polyphyly in Andean tigrinas. (4) Within Tigrina G1, we detected two sub-clusters, one composed of specimens mainly sampled at the Eastern Colombian Andean cordillera and another composed of specimens sampled in the Central and Western Colombian Andean cordilleras as well as in the Ecuadorian Andean cordilleras. The first sub-cluster should be named *Leopardus pardinoides pardinoides* following de Oliveira et al. [71]. The second sub-cluster should be *Leopardus pardinoides oncilla*. With this new result, the proposal of Marín-Puerta [24] that the two Colombian Andean tigrina analyzed by Astorquiza et al. [23] could be classified as *Leopardus pardinoides pardinoides* is unfeasible. *Leopardus pardinoides pardinoides* would only be intended for the tigrinas from the Eastern Andean Colombian cordillera. Additionally, a new nomenclature (scientific name or binomen) is required for Tigrina G2. This group of tigrinas is molecularly highly differentiated from Tigrina G1 and is dispersed in different areas of Colombia and Ecuador, although its morphology is not clearly differentiated from Tigrina G1. Another possibility is that Tigrina G2 may be the traditional *L. tigrinus tigrinus* from Venezuela, Guyana, and northern Brazil that has penetrated the Andes. (5) The BAPS analyses revealed that the differentiation of the Nariño cat from other groups depends on which species or taxa were included in these analyses. It is very important to compare the Nariño cat, as well as all the “rare” tigrinas, with the maximum number possible of specimens of all the existent taxa of the *Leopardus* genus to infer its real systematic position. (6) We revised some of the multiple potential geographical and ecological barriers, which could be responsible for the possible systematic division of Andean tigrina. (7) We also showed the extremely complex and confusing taxonomy of tigrinas throughout history since this taxon was first defined. The new molecular results will surely not simplify this issue. (8) To obtain the binomen *L. narinensis*, we acted in good faith, although some temporary and unforeseeable events that overtook us did not occur in the most orthodox way possible. Likewise, for most new tigrina taxa, binomen is not the best choice and it may be better to talk about lineages with common names or distinctive numbering. (9) Although criticism is essential in science, many of the claims made by Astorquiza et al. [23] and Marín-Puerta [24] are biased, imprecise, and some of them inconsistent. (10) The most realistic method to try to understand and solve the evolutionary and systematic problem of tigrinas is to carry out in-depth studies with WGS.

## Figures and Tables

**Figure 1 animals-15-01891-f001:**
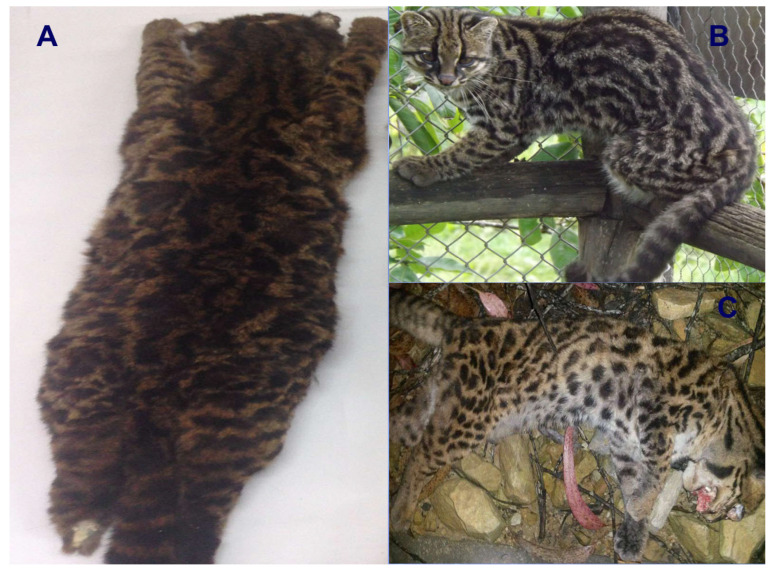
Some of these specimens that individually were genetically compared to the Nariño cat. All of them were morphologically different to the Nariño cat and all of them were morphologically different among them. (**A**) Specimen of *Leopardus tigrinus* from Reserva Forestal Bremen, Municipality of Circasia, Quindío department, Colombia; a female obtained at 1900 m above sea level (masl), IAVH 7331. (**B**) Specimen of *L. tigrinus* from Intag, Imbabura province in northern Ecuador sampled at 3000 masl. (**C**) Specimen of *L. tigrinus* found run over in the road Bogotá-La Calera (Colombia) at 2800 masl.

**Figure 2 animals-15-01891-f002:**
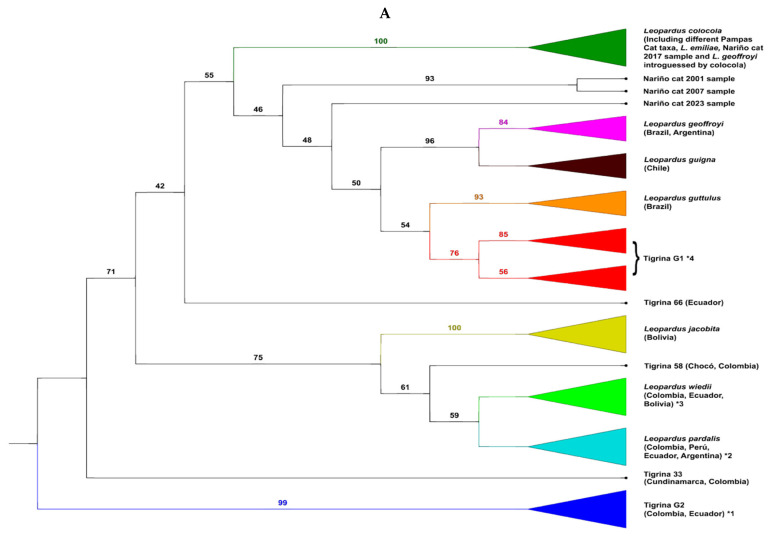
Phylogenetics trees obtained to determine the evolutionary relationships between the tigrinas and the other species of the genus *Leopardus*. (**A**) Maximum Likelihood (ML) tree based off mt*ND5* gene data (total of 164 specimens of all the species of *Leopardus*, including 63 tigrinas, and four temporal replicates from the Nariño cat). *^1^ Group of Tigrina G2; *^2^ Group of ocelots, which enclosed some introgressed and/or hybridized tigrinas; *^3^ Group of margays, which enclosed some introgressed and/or hybridized tigrinas; *^4^ Group of Tigrina G1. (**B**) Neighbor-joining (NJ) tree derived from mt*ND5* data (total of 164 specimens of all the species of *Leopardus*, including 63 tigrinas, and four temporal replicates from the Nariño cat). *^1^ Group of Tigrina G2; *^2^ Group of ocelots, which enclosed some introgressed and/or hybridized tigrinas; *^3^ Group of margays, which enclosed some introgressed and/or hybridized tigrinas; *^4^ Group of Tigrina G1. (**C**) ML tree derived from the complete mitogenome data (total of 102 specimens of all the species of *Leopardus*, including 42 tigrinas, and three temporal replicates from the Nariño cat). *^1^ Group of Tigrina G2; *^2^ Group of Tigrina G1; *^3^ Group of margays, which enclosed some introgressed and/or hybridized tigrinas; *^4^ Group of ocelots, which enclosed some introgressed and/or hybridized tigrinas. (**D**) NJ tree based off complete mitogenome data (total of 102 specimens of all the species of *Leopardus*, including 42 tigrinas, and three temporal replicates from the Nariño cat). *^1^ Group of Tigrina G1; *^2^ Group of Tigrina G2; *^3^ Group of margays, which enclosed some introgressed and/or hybridized tigrinas; *^4^ Group of ocelots, which enclosed some introgressed and/or hybridized tigrinas. In nodes, bootstrap percentages.

**Figure 3 animals-15-01891-f003:**
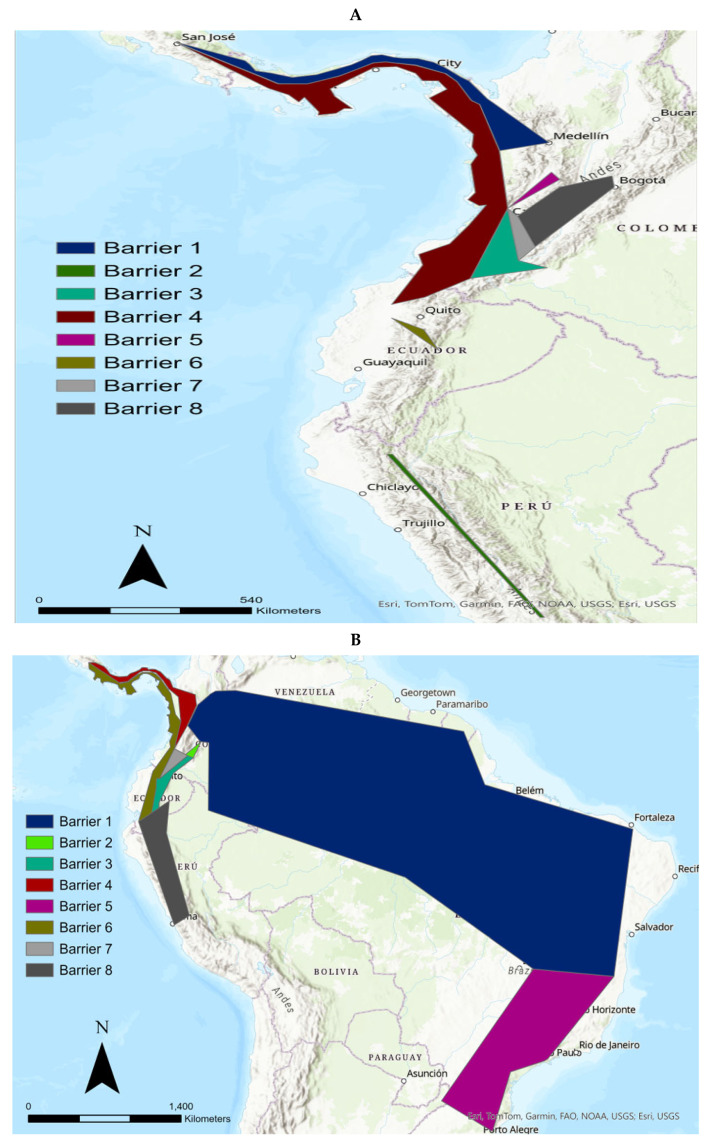
(**A**) Monmonier Algorithm (MA) applied to the mt*ND5* data of the tigrinas (excluding the tigrinas introgressed with margay and ocelot, and *L. guttulus*, and *L. t. emiliae*). Eight potential geographical barriers were determined for these tigrinas. (**B**) MA procedure for the overall mitogenomes of the tigrinas studied (in this case including *L. guttulus* and *L. t. emiliae*).

**Figure 4 animals-15-01891-f004:**
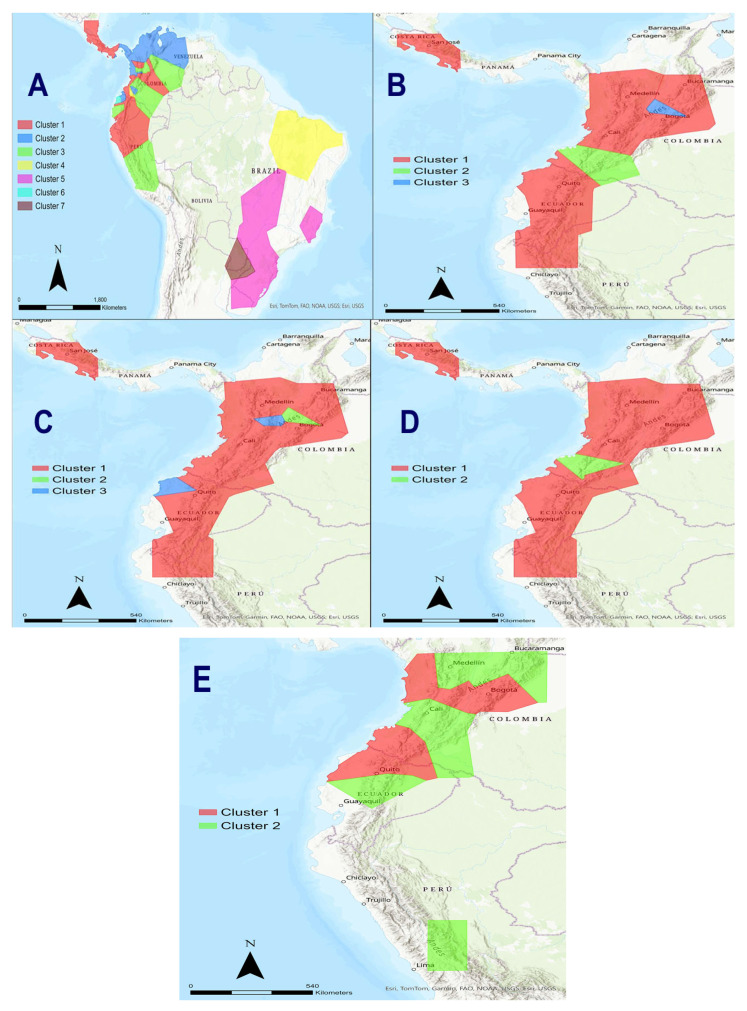
Application of the Bayesian Analysis of Population Structure (BAPS) to determine how many different groups of tigrinas were detected in our molecular data. (**A**) In a first strategy, only the mt*ND5* gene with 63 specimens were considered (including all 37 northern tigrinas analyzed irrespective of their positions in the phylogenetic trees, the four temporal samples of the Nariño cat, nine *L. t. emiliae*, and 13 *L. guttulus*). (**B**) In a second strategy, also with the mt*ND5*, 21 tigrinas were used belonging “a priori” to Tigrina G1 (17 tigrinas plus the four temporal Nariño cat samples). (**C**) A third strategy considered the complete mitogenomes with 22 tigrinas, including those previously observed in the phylogenetic trees as Tigrina G1 and those placed in a unique branch of the phylogenetic trees (tigrinas introgressed/hybridized with ocelot and margays as well as Pampas cats, *L. t. emiliae*, and *L. guttulus* were not included). (**D**) A fourth strategy focused on the complete mitogenomes of 18 tigrina (Tigrina G1 and the three temporal Nariño cat samples [the other “strange” tigrinas were excluded]). (**E**) A fifth strategy considered the complete mitogenomes of 17 tigrinas (Tigrina G2 and all the rare tigrinas in a unique branch in the phylogenetic trees, and the three temporal Nariño cat samples).

**Figure 5 animals-15-01891-f005:**
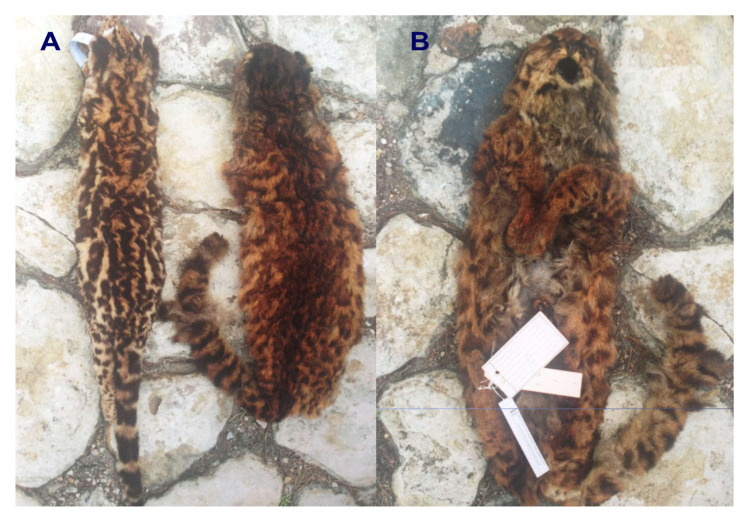
Different aspects and comparisons of the skin of the Nariño cat throughout the time and with other specimens. (**A**) Comparison of the skin of the Nariño cat with a specimen female of tigrina obtained in the Caquetá department (Colombia) obtained in 1972, IAVH0783. This last specimen belonged to the group Tigrina G2. This photo was taken in 2001. (**B**) Ventral view of the Nariño cat taken in 2001. The Nariño cat has a reddish entire ventral area. On the other hand, “traditional” tigrinas have white ventral fur. (**C**) Comparison between the Nariño cat and a skin of a specimen of *Leopardus tigrinus* from Reserva Forestal Bremen, Municipality of Circasia, Quindío department, Colombia. Its phylogenetic position is variable depending on the analyses carried out. (**D**) Comparison between the Nariño cat and a skin of a specimen of *L. tigrinus* from Vrda. Capilla, Cañón Mamarramos, Santuario de Flora y Fauna Iguaque, Villa de Leyva, Boyacá department, Colombia, IAVH 8608. This specimen was obtained in 2011 at 2800 masl. This last specimen belonged to the group of Tigrina G1. The photo was taken in 2023. (**E**) Comparison of the Nariño cat with two other skins of specimens of *L. tigrinus*. In the middle of the photo, the skin of a tigrina from Tapias River, Cuchilla el Guayabo, Vrda. La Cristalina, Municipality of Neira, Caldas department, Colombia, obtained in 2003 at 3300 masl, IAVH 7343. At the left, a skin of a specimen of tigrina from the Municipality of Coper, Boyacá department, Colombia, obtained in 1970 at 950 masl, IAVH 1781. Both last specimens belonged to the group Tigrina G2. The photo was taken in 2023. In 2001 and in 2017, the skin of the Nariño cat was in perfect condition (**A**,**B**). Its hue was a deep red with a very dark hood that extended down the middle of the back. Although 16 years passed between samplings, the skin was intact and had the same appearance. However, when we returned to sample the skin in 2023 (six years later; **C**,**D**), its appearance was noticeable deteriorated. As can be seen, its reddish hue decreased considerably, its dark top also loss of intensity, the rosettes lost shape and hue, and the quality of the coat was damaged. However, despite this deterioration, the skin can be perfectly distinguished from the skins of other tigrinas.

**Table 1 animals-15-01891-t001:** (**A**) Kimura 2P genetic distances (in %) at the mt*ND5* gene between the Nariño cat sequences obtained in different years (2001, 2007, 2017, 2023) and other selected tigrina specimens, which have different morphological traits and from diverse geographical areas and other selected species of the *Leopardus* genus (different subspecies of *Leopardus colocola*, *Leopardus guigna*, *Leopardus geoffroyi*. (**B**). Kimura 2P genetic distances (in %) for complete mitogenomes between the Nariño cat sequences obtained in different years (2001, 2017, 2023) and other selected tigrina specimens, which have different morphological traits and from diverse geographical areas and other selected species of the *Leopardus* genus (different subspecies of *Leopardus colocola*, *Leopardus guigna*, *Leopardus geoffroyi*.

**A**							
**Comparisons**	**Nariño Cat 2001**		**Nariño Cat 2007**		**Nariño Cat 2017**		**Nariño Cat 2023**
Different temporal samples of the Nariño cat	With 2007 sample: 0.5 ± 0.5; with 2017 sample: 5.1 ± 1.4; with 2023 sample: 2.0 ± 0.8		With 2017 sample: 5.9 ± 0.5; with 2023 sample: 2.0 ± 0.8		With 2023 sample: 5.6 ± 1.5		-------------------
Tigrina from Circasia, Quindío, Colombia	6.0 ± 1.5		5.2 ± 1.4		9.4 ± 2.1		3.8 ± 1.2
Tigrina run over on the northern highway in the outskirts of Bogotá, Colombia	4.4 ± 1.3		3.7 ± 1.2		7.6 ± 1.8		2.5 ± 0.9
Tigrina run over in the road Bogotá- La Calera, Colombia	4.4 ± 1.3		3.7 ± 1.2		7.6 ± 1.8		2.3 ± 0.9
Tigrina from Astorquiza et al. [23], Caldas, Colombia	7.1 ± 1.7		6.3 ± 1.6		10.7 ± 2.3		4.9 ± 1.4
Tigrina from Astorquiza et al. [23], Antioquia, Colombia	6.7 ± 1.7		5.9 ± 1.5		10.2 ± 2.2		4.5 ± 1.3
Tigrina from the Nariño department, Colombia, but physically different to the Nariño cat	5.6 ± 1.5		4.8 ± 1.4		8.9 ± 2.0		3.4 ± 1.1
Tigrina from Intag, Imbabura, Ecuador	6.3 ± 1.6		5.5 ± 1.5		9.8 ± 2.2		4.1 ± 1.2
Two tigrinas from Costa Rica, Central America	7.5 ± 1.5		6.7 ± 1.7		10.2 ± 2.2		6.0 ± 1.6
*Leopardus colocola braccatus*	5.5 ± 1.5		6.3 ± 1.6		2.3 ± 0.9		6.0 ± 1.6
*Leopardus colocola steinbachi*	5.2 ± 1.4		5.9 ± 1.5		2.0 ± 0.8		5.6 ± 1.5
*Leopardus guigna*	7.5 ± 1.8		7.5 ± 1.8		10.2 ± 2.2		6.0 ± 1.6
Two *Leopardus geoffroyi*	7.5 ± 1.8 and 9.2 ± 2.0		7.5 ± 1.9 and 8.8 ± 2.0		9.7 ± 2.1		6.0 ± 1.6 and 7.2 ± 1.8
**B**							
**Comparisons**		**Nariño cat 2001**		**Nariño cat 2017**		**Nariño cat 2023**	
Different temporal samples of the Nariño cat		With 2017 sample: 5.4 ± 0.009; with 2023 sample: 4.4 ± 0.008		With 2023 sample: 2.3 ± 0.005		-------------------	
Tigrina from Circasia, Quindío, Colombia		5.2 ± 0.009		5.7 ± 0.009		3.9 ± 0.007	
Tigrina run over on the northern highway in the outskirts of Bogotá, Colombia		5.0 ± 0.008		2.7 ± 0.006		1.1 ± 0.004	
Tigrina run over in the road Bogotá- La Calera, Colombia		5.0 ± 0.008		2.7 ± 0.006		1.1 ± 0.004	
Tigrina from Astorquiza et al. [23], Caldas, Colombia		6.2 ± 0.010		3.7 ± 0.007		2.0 ± 0.005	
Tigrina from Astorquiza et al. [23], Antioquia, Colombia		6.2 ± 0.010		3.7 ± 0.007		1.8 ± 0.005	
Tigrina from the Nariño department, Colombia, but physically different to the Nariño cat		6.2 ± 0.010		4.0 ± 0.007		2.3 ± 0.005	
Tigrina from Intag, Imbabura, Ecuador		5.5 ± 0.009		5.8 ± 0.009		4.0 ± 0.007	
Two tigrinas from Costa Rica, Central America		6.2 ± 0.010		4.3 ± 0.008		3.1 ± 0.006	
*Leopardus colocola braccatus*		6.4 ± 0.010		4.1 ± 0.008		5.9 ± 0.009	
*Leopardus colocola steinbachi*		6.1 ± 0.009		4.1 ± 0.008		5.9 ± 0.009	
*Leopardus guigna*		6.4 ± 0.010		5.9 ± 0.009		4.9 ± 0.008	
*Leopardus geoffroyi*		6.2 ± 0.010		5.6 ± 0.009		5.1 ± 0.009	

**Table 2 animals-15-01891-t002:** Kimura 2P genetic distances (in %) at the mt*ND5* gene for all the selected tigrina pairs that in Table 1 were compared with the Nariño cat below the diagonal. Kimura 2P genetic distances (in %) for the total mitogenomes for all the selected tigrina pairs that in Table 1 were compared with the Nariño cat above the diagonal.

Comparisons	Tigrina from Circasia, Quindío, Colombia	Tigrina Run over on the Northern Highway in the Outskirts of Bogotá, Colombia	Tigrina Run over in the Road Bogotá- La Calera, Colombia	Tigrina from Astorquiza et al. [23], Caldas, Colombia	Tigrina from Astorquiza et al. [23], Antioquia, Colombia	Tigrina from the Nariño Department, Colombia, but Physically Different to the Nariño Cat	Tigrina from Intag, Imbabura, Ecuador	Two tigrinas from Costa Rica, Central America
Tigrina from Circasia, Quindío, Colombia	-	2.7 ± 0.006	2.7 ± 0.006	3.2 ± 0.007	3.2 ± 0.007	3.2 ± 0.007	3.6 ± 0.007	4.0 ± 0.007
Tigrina run over on the northern highway in the outskirts of Bogotá, Colombia	1.3 ± 0.7	-	0.0 ± 0.0	0.9 ± 0.003	0.9 ± 0.003	1.2 ± 0.004	2.8 ± 0.006	1.9 ± 0.005
Tigrina run over in the road Bogotá- La Calera, Colombia	1.3 ± 0.7	0.0 ± 0.0	-	0.9 ± 0.003	0.9 ± 0.003	1.2 ± 0.004	2.8 ± 0.006	1.9 ± 0.005
Tigrina from Astorquiza et al. [23], Caldas, Colombia	2.3 ± 0.9	2.3 ± 0.9	2.3 ± 0.9	-	0.22 ± 0.002	1.7 ± 0.005	3.3 ± 0.007	1.7 ± 0.005
Tigrina from Astorquiza et al. [23], Antioquia, Colombia	2.0 ± 0.8	2.0 ± 0.8	2.0 ± 0.8	0.3 ± 0.3	-	1.7 ± 0.005	3.3 ± 0.007	1.7 ± 0.005
Tigrina from the Nariño department, Colombia, but physically different to the Nariño cat	1.0 ± 0.6	1.0 ± 0.6	1.0 ± 0.6	2.0 ± 0.8	1.6 ± 0.7	-	2.5 ± 0.006	2.2 ± 0.005
Tigrina from Intag, Imbabura, Ecuador	1.6 ± 0.7	1.6 ± 0.7	1.6 ± 0.7	2.7 ± 1.0	2.3 ± 0.9	0.6 ± 0.5	-	3.9 ± 0.007
Two tigrinas from Costa Rica, Central America	3.4 ± 1.1	3.4 ± 1.1	3.4 ± 1.1	2.3 ± 0.7	2.0 ± 0.8	3.0 ± 1.1	3.7 ± 1.2	-

**Table 3 animals-15-01891-t003:** Kimura 2P genetic distances (in %) between the consensus sequence for the four sequences of the Nariño cat obtained in 2001, 2007, 2017, 2023 at the mt*ND5* and different *Leopardus* taxa, and between the consensus sequence for the three sequences of the Nariño cat obtained in 2001, 2017, 2023 for complete mitogenome and different *Leopardus* taxa. For *Leopardus colocola garleppi*, no mitogenome data was obtained.

Compared with Consensus Sequence of the Nariño Cat	mt*ND5* Gene	Complete Mitogenomes
Tigrina G1	3.2 ± 1.0	1.3 ± 0.005
*Leopardus colocola braccatus*	2.6 ± 0.8	4.2 ± 0.006
*Leopardus colocola steinbachi*	2.9 ± 0.9	4.1 ± 0.005
*Leopardus colocola garleppi*	3.4 ± 1.0	--------------
*Leopardus guigna*	6.1 ± 1.4	4.2 ± 0.008
*Leopardus geoffroyi*	6.2 ± 1.3	3.8 ± 0.007
*Leopardus geoffroyi* introgressed by *L. colocola* in Bolivia and in Argentina	3.2 ± 0.9	2.8 ± 0.004

## Data Availability

The datasets generated and analyzed during the current study are available from the corresponding author on reasonable request at the e-mails mruizgar@yahoo.es and mruiz@javeriana.edu.co. The GenBank accession numbers are as follows: PQ579121-PQ579150; OR725153-OR725174; MG230196-MG230233; OR522799-OR522799; OR557277-OR557296; PQ164764-PQ164774; OL615026-OL615080; MG230238-MG230242; MN005818-MN005921; MG23042-MG23045; OR725175-OR725177; OR673479-OR673494; OR520786-OR520799; OR685121-OR685121; MG230236-MG230237.

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
