# Peer review of "The Nariño Cat, the Tigrinas and Their Problematic Systematics and Phylogeography: The Real Story"

_animals, 2025, doi:10.3390/ani15131891_

Round 1

Reviewer 1 Report

Comments and Suggestions for Authors

Manuel et al studied phylogenetics and systematics of genus Leopardus.  The main aims of the their work are: (1) To clarify whether or not the temporal samples obtained of the Nariño cat (four samples at the mtND5 gene, and three at the mitogenomes) show the same molecular and phylogenetic results and if they don’t to provide an explanation as to why they are different, and (2) To present evidence (refuting claims by Astorquiza et al. and Lescroart et al. by studying very few specimens) that the Andean tigrinas are made up of multiple polyphyletic lineages.

The manuscript has high significance of species conservation and management. However, some issues need to be addressed that are lacking in submitted manuscript:

Contamination Issues: While contamination is plausible, the hypothesis lacks direct experimental validation (e.g., replication in independent labs).

Nuclear DNA Absence: Reliance on mitochondrial data limits insights into nuclear introgression; integrating nuclear markers would strengthen claims.

Morphological Data: Limited quantitative morphological comparisons weaken the case for the Nariño cat’s distinctiveness.

Author Response

In red, all the changes that there was added to the new version of the manuscript. Extensive changes have been introduced especially in the Discussion (following the comments of Referee 2). The answers to Referee 1 are in the uploaded file in this section. Now, a new sections of simple summary and conclusions were added to this new manuscript version. The Javier Vega CV is sending you via e-mail.

Reviewer 2 Report

Comments and Suggestions for Authors

Ruiz-García et al. (“The Nariño Cat, the Tigrinas and Their Problematic Systematic Positions (Leopardus, Felidae, Carnivora): The Real Story”) present mitochondrial data analyses from Nariño cat specimens (2001, 2007, 2017, 2023) and extensive sequences for other Leopardus samples. They confirm eight divergent tigrina clades, highlight a previously unreported northern Andean group, and argue that only the 2001/2007 Nariño tissues are uncontaminated, reinforcing the distinctiveness of Leopardus narinensis.

The analytical design and laboratory procedures are sound and qualify for publication consideration.

However, because the taxonomic conclusions remain controversial, the manuscript would benefit from broader engagement with recent literature and a reframing emphasizing phylogeography and systematics rather than delivering a conclusive taxonomic verdict.

  • The manuscript cites no sources post-2023. Key contributions such as the thesis of Sebastián Marín Puerta (2024) “Leopardus narinensis Ruiz-García, Pinedo-Castroy Shostell, 2023, es un sinónimo menor de Leopardus pardinoides (Gray 1867): La necesidad de buenas prácticas en las propuestas nomenclaturales y taxonómicas” that directly challenge the species status of narinensis should be discussed plus other 2024–2025 papers to situate the claims within the evolving consensus or lack thereof on Leopardus taxonomy.
  • Where disagreements exist (e.g., synonymy with L. pardinoides), articulate the evidence on both sides (morphology, genomics, nomenclatural best practice) and explain how the present results support or contradict them.
  • Current title and narrative foreground a taxonomic defence of L. narinensis. Given the persisting debate, consider repositioning the study as a broader investigation of tigrina phylogeography, genomic diversity, and historical biogeography. A possible revised title: “The Nariño Cat, the Tigrinas, and Their Problematic Systematics and Phylogeography,” then restructure the discussion to defer definitive taxonomic conclusions.
  • The study stresses genetic data; however, morphology, pelage patterning, cranial metrics, and ecology remain decisive in felid species delimitation. Either include at least a synopsis of existing morphometric comparisons or acknowledge this limitation.
  • Acknowledge recommendations on nomenclatural best practice (code compliance, designation of type material, diagnostic criteria), addressing whether the original L. narinensis description meets ICZN standards in light of new evidence.
  • Tone down narrative elements about the “discovery story” for a chronological, evidence-based account of Leopardus species concepts since Gray 1867.

With these revisions, particularly wider literature integration, explicit acknowledgement of taxonomic uncertainties, and a shift toward a systematics + phylogeography perspective, the manuscript will more effectively advance understanding of Leopardus diversification while embracing the cumulative, iterative nature of taxonomic science.

Author Response

In red, all the changes that there was added to the new version of the manuscript. Extensive changes have been introduced especially in the Discussion (following the comments of Referee 2). The answers to Referee 2 are in the uploaded file in this section. Now, a new sections of simple summary and conclusions were added to this new manuscript version. The Javier Vega CV is sending you via e-mail.

Reviewer 3 Report

Comments and Suggestions for Authors

The paper by Ruiz-García et al. addresses a putative new species within the Leopardus genus. This is a compelling topic, especially considering that there may still be undiscovered species. However, several issues with the data need to be resolved. Given the paper's length, it is crucial that it be accessible to readers, so I will begin with the most significant points.

Firstly, it should be clearly stated that the term “temporal Nariño cat samples [2001, 2007, 2017, 2023]” refers to samples taken from a single specimen across different years. Initially, I interpreted “temporal Nariño cat samples” as attempts to collect multiple animals in different years, and only by the end of the paper did I grasp the actual meaning.

The phylogenetic trees show four sequences (2001, 2007, 2017, 2023). The authors suggest that the 2017 and 2023 samples were likely contaminated with DNA from either the Pampas cat or Tigrina G1, while asserting that the 2001 and 2007 samples belong to the bona fide Nariño cat. If that is the case, why are the sequences from 2001 and 2007 not identical? Additionally, why do the 2017 and 2023 samples occupy distinct positions on the tree, and not within the respective clades of the Pampas cat and Tigrina G1? They look like they also represent distinct new species. Regarding the analysis of the complete mitochondrial genomes, chimeric regions can result from the erroneous assemnly of mtDNA amplicons from different species, leading to conflicting phylogenetic placements. However, it is unclear how this could occur with short ND5 fragments. This issue is perplexing and needs clarification. Bioinformatic methods, such as sliding window analysis, could identify chimeric regions. Anyway, submitting questionable sequences to GenBank is a bad practice, as they may be treated as valid sequences from real animals in future analyses of other authors.

The Introduction provides an extensive overview of the species in question. However, in section 2.1 (Samples), it is difficult to discern which specimens were sequenced and which were sourced from previous studies. This should be made explicit: “We sequenced the following specimens: … Data for specimens … were taken from GenBank [references].”

Regarding Table 1: why are the distances presented with errors (e.g., 0.6 + 0.5) if each sample contains only one sequence? Additionally, the values such as “Between 2001-2007: 0.6 + 0.5; between 2001-2017: 5.1 + 1.4; between 2001-2023: 2.0 + 0.8” are confusing. I recommend splitting this row into three for clarity.

I find Table 3 problematic. The authors indicate that the specimens from 2001, 2007, 2017, and 2023 belong to different lineages (possibly species). However, they then create a consensus and compare it to different species. The rationale for analyzing such a non-natural consensus is unclear.

Furthermore, the Neighbor-Joining (NJ) algorithm is quite outdated; contemporary studies typically employ Maximum Likelihood (ML), Maximum Parsimony (MP), or Bayesian analysis.

Regarding the study's conclusions: the evidence for cryptic lineages is significantly weakened by the presence of contaminating/chimeric sequences. And the proof of the existence of new species is even more doubtful: the authors themselves note numerous instances of mtDNA introgression, making it impossible to determine the number of species based solely on mtDNA lineages. I believe that Whole Genome Sequencing (WGS) of the specimen in question will eventually clarify these issues.

Author Response

In red, all the changes that there was added to the new version of the manuscript. Extensive changes have been introduced especially in the Discussion (following the comments of Referee 2). The answers to Referee 3 are in the uploaded file in this section. Now, a new sections of simple summary and conclusions were added to this new manuscript version. The Javier Vega CV is sending you via e-mail.

Round 2

Reviewer 1 Report

Comments and Suggestions for Authors

No further comments

Reviewer 3 Report

Comments and Suggestions for Authors

I have reexamined the manuscript, and found the authors have made substantial revisions to the text. Although I still have some disagreements on certain points, I believe the paper has been improved sufficiently to be accepted.